# MEMORY EFFICIENT FINE-TUNING OF LLMS VIA FORWARD-ONLY HESSIAN-FREE COORDINATE DESCENT

## ABSTRACT

Fine-tuning large language models (LLMs) for specific downstream tasks has traditionally relied on memory-intensive optimizers using classical backpropagation, which demands substantial memory to store model states for gradient computation, motivating the development of memory-efficient zeroth-order optimizers that operate in a forward-only manner. However, the slower convergence of the zeroth-order optimizer remains a challenge, which recent research addresses by incorporating Hessian information to accelerate training, although storing even the diagonal Hessian requires memory equivalent to that of the model weights, leading to significant memory usage. To mitigate this problem, we propose a zeroth-order block coordinate descent (BCD)-Newton optimizer with coordinate updates adaptive to second-order information, allowing us to treat model layers as separate blocks and update only a greedily selected subset per training iteration, thereby reducing memory requirements while accelerating convergence. Specifically, at each iteration, an active set of layers is selected according to the block Gauss-Southwell-Diagonal rule, and their weights are updated while the other layers remain fixed, with compressed diagonal Hessian information stored and updated exclusively for the active layers. For fine-tuning foundation models across small to large sizes (OPT-1.3B and 30B, LLaMA-2-7B), our method achieves up to 40% memory reduction compared to existing Hessian-informed zeroth-order methods, while preserving baseline accuracy and memory usage to zeroth-order methods across various tasks, offering a memory-efficient alternative method for LLMs fine-tuning, especially on memory-constrained devices.

## 1 INTRODUCTION

Fine-tuning transformer-based large models is essential in adapting pre-trained models to specific downstream tasks and improving performance (Raffel et al., 2020). This process also allows the model to continue training on lower-end devices compared to pre-training, improving accessibility and reducing the training cost. To achieve these goals, parameter-efficient fine-tuning (PEFT) techniques like LoRA (Hu et al., 2021) have been proposed to enable fine-tuning on consumer-level GPUs or edge devices, providing significant economic and practical benefits. Typically, fine-tuning employs traditional optimizers like SGD or Adam (Kingma, 2014), and updates model weights within the backpropagation process. However, this approach requires storing parameters, gradients, activations, and optimizer states, resulting in substantial memory overhead (Lv et al., 2023b;a; Rajbhandari et al., 2020). As increased model sizes and larger batch sizes are employed for training, the memory demands of traditional optimizers have become a significant bottleneck for devices with limited memory resources, even when PEFT methods are applied (Cai et al., 2020). To address this challenge, our work explores memory-efficient techniques further to reduce the memory overhead during fine-tuning on low-end devices.

To tackle the memory inefficiency issue, recent advancements have explored the use of zeroth-order optimizers such as MeZO (Malladi et al., 2023) that estimate the gradients with only forward passes, which eliminates the need for backpropagation, thereby significantly reducing memory consumption by avoiding the storage of intermediate optimizer states. Though memory-efficient, the slower convergence rates of zeroth-order optimizers have limited their practical utility. To accelerate

convergence, researchers have incorporated second-order information, such as diagonal Hessian approximations as in HiZOO (Zhao et al., 2024b), into the optimization process. However, these solutions can introduce significant memory overhead, as incorporating second-order information often incurs substantial memory cost.

To efficiently and effectively utilize second-order information to tackle the convergence challenge of zeroth-order methods, inspired by recent advances in efficient BCD rules and updates (Nutini et al., 2017) and in applying BCD with Adam (Kingma, 2014) or AdamW Loshchilov (2017) optimizations for training LLMs (Pan et al., 2024; Luo et al., 2024), we introduce a layerwise BCD scheme to optimize memory usage and convergence, treating model layers as independent blocks and selectively activating a subset of layers during each iteration based on their tracked saliency scores derived from the block Gauss-Southwell-Diagonal (GSD) rule (Nutini et al., 2017). In practice, to mitigate the computational overhead, we employ a bandit method for efficient rule-based layer selection sampling, further improving computational efficiency to achieve optimal training performance. This approach enhances convergence by focusing on the most influential layers and reducing the memory and computational overhead of convergence-focused methods.

Through extensive experiments on a single NVIDIA RTX 4090 GPU or RTX A6000 GPUs, we demonstrate that our proposed method enhances training efficiency and memory management while fine-tuning foundation models, including OPT-1.3B and 30B(Zhang et al., 2022b) and LLaMA-2-7B (Touvron et al., 2023). As a forward-only training pipeline with coordinate updates driven by the GSD rule, it empirically delivers superior convergence speed and accuracy compared to MeZO. Additionally, compared to the HiZOO baseline, our approach achieves comparable or better performance with reduced computational and memory overhead across multiple GLUE (Wang, 2018) and SuperGLUE (Wang et al., 2019) tasks. These improvements make our method particularly well-suited for fine-tuning large models on devices with limited memory, expanding the accessibility of large language models in real-world applications.

In summary, our main contributions are three-fold:

- We propose a novel zeroth-order BCD-Newton optimizer with coordinate updates driven by Gauss-Southwell rules to address the convergence issue in zeroth-order optimization, offering a practical and convergence-enhanced alternative to MeZO.

- We design a bandit method derived from the GSD rule to perform probabilistic sampling for efficient layer selection, which reduces the computational overhead of convergence-focused methods. By adaptively updating weights across coordinates of the model layers based on their rule-based Hessian-aware saliency score, this method manages blockwise updates efficiently and improves convergence. In addition, we establish the convergence analysis for the proposed algorithm.

- We conduct experiments on fine-tuning OPT-1.3B/30B and LLaMA-2-7B across various settings, demonstrating that our proposed method achieves comparable or better performance with reduced computational overhead compared to strong baselines.

## 2 RELATED WORK

**First and second order optimization for LLMs.** Traditional first-order optimizers, such as SGD, AdaGrad (Duchi et al., 2011), and RMSProp (Tieleman et al., 2012), are foundational tools in deep learning. Adam (Kingma, 2014), with its adaptive moment estimates for faster convergence, and its variant AdamW (Loshchilov, 2017), which modifies the weight decay term to improve generalization, have become the dominant optimizers for fine-tuning LLMs. Second-order optimization methods incorporating Hessian information, such as K-FAC (Martens & Grosse, 2015), EVA (Zhang et al., 2022a), AdaHessian (Yao et al., 2021), and Sophia (Liu et al., 2023), have been explored to accelerate convergence further. However, estimating the Hessian is computationally and memory-intensive, particularly with the growing size of LLMs, making most second-order methods not practical for fine-tuning on devices with memory constraints.

**Zeroth-order (ZO) Optimization.** A classical zeroth-order optimization method, SPSA (Spall, 1992), estimates the gradient using two forward passes before and after parameter perturbation. Recently, MeZO (Malladi et al., 2023) adapted SPSA's corresponding SGD variant ZO-SGD by

using a random number generator, enabling an in-place implementation that significantly reduces memory usage for storing random vectors during training. Based on MeZO, recent work explores its variants like incorporating sparsity for memory efficiency (Guo et al., 2024) or linear interpolation for more accurate gradient estimates (Wang et al., 2024). Additionally, Zhang et al. (2024) conducted a benchmark study to analyze and enhance zeroth-order fine-tuning methods. However, the convergence performance of ZO methods often falls behind that of first-order methods. To improve convergence, HiZOO (Zhao et al., 2024b) proposed to utilize Hessian information through diagonal Hessian estimation. Beyond these approaches, several other gradient-free methods have been proposed, such as using evolutionary algorithms for gradient-free optimization (Sun et al., 2022b;a).

**Memory-efficient Fine-tuning for LLMs.** Numerous algorithms have been developed to reduce memory costs for training LLMs. Based on backpropagation, practical techniques such as gradient checkpointing (Chen et al., 2016) recompute gradients, FlashAttention (Dao et al., 2022) employs tiling and recomputation to leverage cache for improved efficiency, and the ZeRO optimizers (Rajbhandari et al., 2020; Ren et al., 2021) enable offloading to manage memory usage effectively. Additionally, researchers have utilized compression and quantization methods to approximate gradients, activations, and other optimizer states, enhancing training performance (Jiang et al., 2022; Li et al., 2024). On another front, methods like LOMO (Lv et al., 2023b;a) fuse gradient updates to accelerate training. One notable approach to fine-tuning is parameter-efficient fine-tuning (PEFT) methods, which include techniques such as Adapters (LoRA) (Hu et al., 2021; Houlsby et al., 2019), prompt tuning (Lester et al., 2021), and selective methods like bias-only fine-tuning (Zaken et al., 2021) and layerwise freezing (Brock et al., 2017). In addition, Zhao et al. (2024a) recently introduced GaLore, which reduces memory costs by projecting gradients into a low-rank compact space.

**Block Coordinate Descent methods for LLM Optimization.** The optimization objective in BCD is minimized along coordinate directions successively (Wright, 2015). The recently proposed BAdam (Luo et al., 2024) showcases the effectiveness of combining BCD with Adam. Similarly, LiSA (Pan et al., 2024) improves performance by selectively updating transformer layers with the AdamW optimizer, outperforming LoRA across tasks on LLaMA-2 (Touvron et al., 2023).

Amid the rapid advancements in efficient training for large models, HiZOO and BAdam are the closest related works to ours. However, our approach distinguishes itself by addressing the memory overhead and slow convergence in two key ways: first, by eliminating the need for backpropagation through novel zeroth-order BCD-Newton optimization, and second, by reducing the memory cost of Hessian-informed methods through a rule-based bandit method, correspondingly with a convergence proof. Unlike PEFT methods, our approach enables full parameter fine-tuning, which has yielded superior performance in various tasks (Ding et al., 2022). Overall, our method offers a complementary optimizer-based solution to improve memory efficiency that can be combined with techniques like compression and system-level approaches.

# 3 REVISITING MEMORY COST: A BCD APPROACH

In this section, we provide a brief overview of how zeroth-order (ZO) and Hessian-informed ZO optimizer methods work by introducing the core concepts of MeZO (Malladi et al., 2023) and HiZOO (Zhao et al., 2024b). Next, we introduce BCD methods such as BAdam (Luo et al., 2024). To ensure consistency, we have adapted the definitions from these works. Finally, we reconsider the memory consumption of these methods and propose our BCD-integrated Newton method optimizer.

## 3.1 PRELIMINARIES OF ZEROTH-ORDER OPTIMIZERS

### 3.1.1 SPSA, ZO-SGD, AND MEZO

Let $\mathcal{L}(\boldsymbol{\theta}; \mathcal{B})$ represent the loss function for training the model with parameters $\boldsymbol{\theta} \in \mathbb{R}^d$ on the minibatch $\mathcal{B}$, omitting the $\mathcal{B}$ for simplicity. The SPSA algorithm (Spall, 1992) perturbs the model using $\boldsymbol{z} \in \mathbb{R}^d$, sampled from $\mathcal{N}(0, \mathbf{I}_d)$, and estimates the gradient on the minibatch as follows:

$$\hat{\nabla}\mathcal{L}(\boldsymbol{\theta}) = \frac{\mathcal{L}(\boldsymbol{\theta} + \mu\boldsymbol{z}) - \mathcal{L}(\boldsymbol{\theta} - \mu\boldsymbol{z})}{2\mu}\boldsymbol{z} \approx \boldsymbol{z}\boldsymbol{z}^\top\nabla\mathcal{L}(\boldsymbol{\theta}) \qquad (1)$$

where $\mu$ is the perturbation scale. The corresponding SPSA optimizer, ZO-SGD, employs two forward passes to estimate the gradients. With learning rate $\eta$, ZO-SGD updates the parameters as

$\boldsymbol{\theta}_{t+1} = \boldsymbol{\theta}_t - \eta \hat{\nabla} \mathcal{L}(\boldsymbol{\theta}; \mathcal{B}_t)$. In this vanilla algorithm, the sampled vector $\boldsymbol{z}$ requires memory equivalent to that of the perturbed weights, resulting in a memory cost that is double the cost of inference.

In contrast, MeZO (Malladi et al., 2023) introduces an in-place implementation using a random number generator. Only a random seed $s$ needs to be sampled and stored at each step, allowing the generator to be reset by $s$ to regenerate the vector $\boldsymbol{z}$. This approach eliminates the need to save the vector, reducing the memory cost to match that of inference.

### 3.1.2 HESSIAN-INFORMED PRECONDITIONER WITH MEZO

To harness second-order information through MeZO for enhanced convergence rates, Zhao et al. (2024b) introduce HiZOO, utilizing a diagonal Hessian-based preconditioner that adjusts the update sizes of parameters based on their curvature. By estimating and storing only the diagonal Hessian, HiZOO requires $\mathcal{O}(d)$ memory, significantly less than the $\mathcal{O}(d^2)$ needed for the full Hessian matrix.

The descent direction is defined as $\boldsymbol{g}_\mu(\boldsymbol{\theta}_t) = \frac{\mathcal{L}(\boldsymbol{\theta}_t + \mu \boldsymbol{\Sigma}_t^{1/2} \boldsymbol{z}) - \mathcal{L}(\boldsymbol{\theta}_t - \mu \boldsymbol{\Sigma}_t^{1/2} \boldsymbol{z})}{2\mu} \boldsymbol{\Sigma}_t^{1/2} \boldsymbol{z}$. Let $\boldsymbol{\Sigma}$ denote the estimated inverse Hessian matrix, approximating the diagonal Hessian as a positive definite matrix, with $\boldsymbol{\Sigma}^{-1} \approx \nabla^2 \mathcal{L}(\boldsymbol{\theta})$. Define $\boldsymbol{\Sigma}_t$ as the estimated Hessian inverse at training step $t$, initialized as $\boldsymbol{\Sigma}_0 = \mathbf{I}_d$. Storing $\boldsymbol{\Sigma}_t$ incurs a memory cost of $\mathcal{O}(d)$, and it is updated at each step.

HiZOO approximates the diagonal Hessian using three forward passes to compute $\mathcal{L}(\boldsymbol{\theta} + \mu \boldsymbol{\Sigma}^{1/2} \boldsymbol{z})$, $\mathcal{L}(\boldsymbol{\theta} - \mu \boldsymbol{\Sigma}^{1/2} \boldsymbol{z})$, and $\mathcal{L}(\boldsymbol{\theta})$. By applying Taylor's expansion, they obtain that:

$$\mathcal{L}(\boldsymbol{\theta} \pm \mu \boldsymbol{\Sigma}^{1/2} \boldsymbol{z}) = \mathcal{L}(\boldsymbol{\theta}) \pm \mu \langle \nabla \mathcal{L}(\boldsymbol{\theta}), \boldsymbol{\Sigma}^{1/2} \boldsymbol{z} \rangle + \frac{\mu^2}{2} \boldsymbol{z}^\top \boldsymbol{\Sigma}^{1/2} \nabla^2 \mathcal{L}(\boldsymbol{\theta}) \boldsymbol{\Sigma}^{1/2} \boldsymbol{z} + \mathcal{O}(\mu^3), \quad (2)$$

the difference $\Delta \mathcal{L}$ is then calculated as:

$$\Delta \mathcal{L} = \mathcal{L}(\boldsymbol{\theta} + \mu \boldsymbol{\Sigma}^{1/2} \boldsymbol{z}) + \mathcal{L}(\boldsymbol{\theta} - \mu \boldsymbol{\Sigma}^{1/2} \boldsymbol{z}) - 2\mathcal{L}(\boldsymbol{\theta}) = \mu^2 \boldsymbol{z}^\top \boldsymbol{\Sigma}^{1/2} \nabla^2 \mathcal{L}(\boldsymbol{\theta}) \boldsymbol{\Sigma}^{1/2} \boldsymbol{z} + \mathcal{O}(\mu^3).$$

Based on Ye (2023), the estimation of the diagonal Hessian $\nabla^2 \mathcal{L}(\boldsymbol{\theta})$ at $\boldsymbol{\theta}$ is:

$$\hat{\boldsymbol{\Sigma}}_t = \frac{\Delta \mathcal{L}}{2\mu^2} \left( \boldsymbol{\Sigma}_t^{-1/2} \boldsymbol{z}_i \boldsymbol{z}_i^\top \boldsymbol{\Sigma}_t^{-1/2} - \boldsymbol{\Sigma}_t^{-1} \right), \qquad \text{EMA}: \boldsymbol{\Sigma}_{t+1}^{-1} = (1 - \alpha_t) \boldsymbol{\Sigma}_t^{-1} + \alpha_t \left| \hat{\boldsymbol{\Sigma}}_t \right|, \quad (3)$$

where, in addition to mitigating noise in the computation, an exponential moving average (EMA) is employed to update the diagonal Hessian estimate, and $\alpha_t$ is a smooth scale. The absolute value $|\hat{\boldsymbol{\Sigma}}_t|$ ensures that all entries remain non-negative. In this manner, HiZOO approximates the diagonal entries of $\nabla^2 \mathcal{L}(\boldsymbol{\theta})$ by $\boldsymbol{\Sigma}_t$, requiring one more forward pass per step compared with MeZO.

### 3.1.3 BLOCK COORDINATE DESCENT

At each iteration, block coordinate descent (BCD) fixes all other parameters and optimizes the objective function over the selected coordinates, resulting in an optimization problem with reduced dimension. A natural block partition organizes transformer layers in ascending order for LLMs. Formally, an ordered block partition $\pi = \{\pi_1, \ldots, \pi_i, \ldots, \pi_D\}$ divides the entire model parameters $\boldsymbol{\theta} \in \mathbb{R}^d$ into $D$ blocks, such that $\boldsymbol{\theta} = \{\boldsymbol{\theta}_{\pi_1}, \ldots, \boldsymbol{\theta}_{\pi_i}, \ldots, \boldsymbol{\theta}_{\pi_D}\}$ with $\boldsymbol{\theta}_{\pi_i} \in \mathbb{R}^{d_i}$ and $\sum_{i=1}^D d_i = d$. Based on BCD, BAdam (Luo et al., 2024) propose to incorporate Adam updates as its inner solver and optimize over only one active block $\boldsymbol{\theta}_{\pi_i}$ at a time while keeping the other inactive blocks fixed. Mathematically, BAdam solves the following subproblem at the $t$-th block-epoch for $i = 1, \ldots, D$ to update the active block $\boldsymbol{\theta}_{\pi_i}$:

$$\boldsymbol{\theta}_{\pi_i}^{t+1} \in \arg \min_{\boldsymbol{\theta}_{\pi_i} \in \mathbb{R}^{d_i}} \mathcal{L}(\boldsymbol{\theta}_{\pi_1}^{t+1}, \ldots, \boldsymbol{\theta}_{\pi_{i-1}}^{t+1}, \boldsymbol{\theta}_{\pi_i}, \boldsymbol{\theta}_{\pi_{i+1}}^t, \ldots, \boldsymbol{\theta}_{\pi_D}^t). \quad (4)$$

This subproblem Equation 4 keeps inactive blocks fixed at their latest values, leading to a significantly lower-dimensional optimization problem compared to $\min_{\boldsymbol{\theta}} \mathcal{L}(\boldsymbol{\theta})$.

### 3.2 REVISITING MEMORY COST FROM THE BCD PERSPECTIVE

**Who consumed my memory?** Second-order methods incorporate a full or diagonal Hessian matrix or its estimation, as a preconditioner to accelerate convergence, but this introduces a significant memory

cost of $\mathcal{O}(d)$, where $d$ denotes the number of model parameters. Large models such as LLaMA-2-7B (Touvron et al., 2023) with $d = 7$ billion parameters require over 14GB of memory storage in FP16 precision. When combined with the memory required for model parameters, this easily exceeds the capacity of consumer-level devices, undermining MeZO's original goal of achieving memory efficiency. Our experiments demonstrate that applying Hessian-based optimization steps significantly increases memory usage, as shown in Table 1. Even though Hessian-aware approaches offer performance improvements, the considerable memory overhead of storing Hessian information becomes a bottleneck, nearly costing at least twice the parameter size. This memory-convergence dilemma leads to a situation where the benefits of second-order methods are outweighed by their heavy memory consumption, limiting their practicality in memory-constrained environments. Furthermore, the memory consumption increases with batch size for both first and second-order methods, intensifying the memory overhead.

Table 1: Experiments of actual GPU memory consumption (m) for various algorithms.

| Device | Model | SGD | BCD | LoRA | MeZO | HiZOO | **ours** FOCUS$_{(1,4)}$ |
|--------|-------|-----|-----|------|------|-------|--------------------------|
| RTX 4090 | OPT-1.3B | 23G | 21G | 11G | 4.4G | 7.5G | $3.6G \leq m \leq 4.6G$ |
| RTX A6000 | LLaMA-2-7B | $> 48$G | 46G | 40G | 31G | $> 48$G | 32G |
| Theoretical average memory | | $3d$ | $d < m < 3d$ | $d < m < 3d$ | $d$ | $2d$ | $\approx d$ |

**How to reduce Hessian memory consumption?** To address this memory-convergence tradeoff, we propose integrating BCD into the zeroth-order Newton optimization. BCD allows us to partition the model into blocks, optimizing only a subset of layers at each iteration while keeping the rest frozen. This approach dramatically reduces the memory required for storing Hessian information. For example, by partitioning the aforementioned LLaMA-2-7B model into $D = 32$ blocks, corresponding to its 32 transformer layers, we reduce the additional memory cost associated with Hessian storage to $\frac{d}{D} \ll d$, bringing it to under 1GB of memory, while also reducing the buffer for residual optimization states, significantly improving memory efficiency while preserving the advantages of second-order optimization. Moreover, MeZO is kept to update the embedding and language modeling head layers, avoiding the instability and overhead often associated with second-order methods.

To validate our analysis, we conducted preliminary experiments (detailed in Section 5) measuring the GPU memory consumption of various optimizers during the fine-tuning of medium-sized language models, specifically OPT-1.3B (Zhang et al., 2022b) on an RTX 4090 (24GB) and LLaMA-2-7B on an RTX A6000 (48GB). As Table 1 and Figure 1 briefly illustrate, incorporating diagonal second-order information increases memory demand by over 70% Notably, the allocated memory includes residual state memory such as temporary buffers and fragments (Rajbhandari et al., 2020), which means the overall memory requirement exceeds that of the parameters alone, resulting in the overall increase short of a full 100%. In contrast, our BCD-integrated method significantly reduces memory consumption, bringing it in line with MeZO while maintaining comparable performance. As we will further demonstrate in Section 5, our proposed method achieves comparable accuracy and offers a practical, memory-efficient alternative to MeZO with the extra Hessian-informed benefits.

**Flexibility in BCD rules.** Beyond the natural block partitioning of model layers updates with cyclic selection (the well-known Gauss-Seidel method), BCD can be adapted with various strategies such as in ascending/descending order, randomized block selection (which fixes a worst case for cyclic CD), or score-based sampling (Luo et al., 2024; Pan et al., 2024). Recently, LiSA (Pan et al., 2024) propose to select layers based on predefined probability values. Nutini et al. (2017) discuss block Gauss-Southwell rules that improve performance further. These rules enhance the overall training process in different optimization scenarios while maintaining memory efficiency.

## 4 METHODOLOGY

### 4.1 BCD-INTEGRATED ZO-NEWTON OPTIMIZER

Motivated by revisiting the second-order memory consumption, we identified a significant bottleneck caused by storing diagonal Hessian estimation, which introduced substantial memory overhead, particularly for large models. Ultimately, to address this memory-convergence tradeoff, we propose a new ZO-BCD-Newton optimizer, termed **F**orward-**O**nly **C**oordinate **U**pdates with **S**econd-order Information (**FOCUS**). Recognizing the transformer model's layerwise structure, we treat each layer

as a block for BCD-Newton optimization. We partition the model into blocks and update only a subset of layers at each iteration, reducing the Hessian storage while maintaining the convergence benefits of second-order methods. Additionally, we update the embedding and language model head layers solely through MeZO optimization to mitigate the instability and overhead typically associated with second-order methods. This results in a more memory-efficient approach for fine-tuning.

Formally, for the current step $T$, the parameter block $\boldsymbol{\theta}_{\pi_b}$ to update can be selected using several types of BCD algorithms. However, different methods vary significantly in memory and computational costs. In practice, the substantial computational and storage overhead of importance-score-based updates led us to implement a bandit method featuring a Gauss-Southwell-induced probabilistic distribution. Empirical results from our experiments confirm its effectiveness. The pseudocode for our proposed algorithm is detailed in Algorithm 1. A concise overview of the foundational concepts of BCD and bandit methods is provided, with a more extensive discussion deferred to the Appendix.

---

**Algorithm 1** Training Pipeline of the proposed method.

1: **Input:** parameters $\boldsymbol{\theta} \in \mathbb{R}^d$, loss function $\mathcal{L}$, perturbation scale $\mu$, learning rate $\eta$, smooth scale $\alpha$
2: **for** $t = 1, \ldots, T$ **do**
3:     Select block $\boldsymbol{\theta}_{\pi_b} \in \boldsymbol{\theta}$ according to the BCD rule
4:     **if** a new block is selected **then**
5:         $\boldsymbol{\Sigma} \leftarrow \mathbf{I}_{|\boldsymbol{\theta}_{\pi_b}|}$ ▷ Diagonal Hessian initialization
6:     **end if**
7:     Freeze other layers
8:     Sample a random seed $s$ ▷ First-time sampling
9:     **for** $\mu_i = 0, +\mu, -2\mu$ **do**
10:         **for** $\boldsymbol{\theta}_i \in \boldsymbol{\theta}_{\pi_b}$ **do**
11:             Sample $\boldsymbol{z} \sim \mathcal{N}_s(0, \mathbf{I}_{|\boldsymbol{\theta}_i|})$
12:             $\boldsymbol{\theta}_i \leftarrow \boldsymbol{\theta}_i + \mu_i \boldsymbol{\Sigma}_t^{1/2} \boldsymbol{z}$ ▷ Perturb In-place
13:         **end for**
14:         $\ell_{\text{sign}(\mu_i)} = \mathcal{L}(\boldsymbol{\theta})$
15:     **end for**
16:     $\texttt{projected\_grad} \leftarrow (\ell_+ - \ell_-)\boldsymbol{\Sigma}_t^{1/2}/2\mu$
17:     $\hat{\boldsymbol{\Sigma}}_t \leftarrow \frac{\Delta\ell}{2\mu^2}\boldsymbol{\Sigma}_{t-1}^{-1/2}\boldsymbol{z}_i\boldsymbol{z}_i^\top\boldsymbol{\Sigma}_{t-1}^{-1/2}$ ▷ Hessian Update
18:     $\boldsymbol{\Sigma}_t^{-1} \leftarrow (1 - \alpha_t)\boldsymbol{\Sigma}_{t-1}^{-1} + \alpha_t \left|\hat{\boldsymbol{\Sigma}}_t\right|$
19:     Reset random number generator with seed $s$
20:     **for** $\boldsymbol{\theta}_i \in \boldsymbol{\theta}_{\pi_b}$ **do**
21:         Sample $\boldsymbol{z} \sim \mathcal{N}_s(0, \mathbf{I}_{|\boldsymbol{\theta}_i|})$
22:         $\boldsymbol{\theta}_i \leftarrow \boldsymbol{\theta}_i - \eta_t * \texttt{projected\_grad} * \boldsymbol{z}$
23:     **end for**
24: **end for**

---

**Algorithm 2** Gauss-Seidel Selection

1: **Input:** $\{\boldsymbol{\theta}_{\pi_i}\}_{i=1}^D$, current step $t$
2: Select $\boldsymbol{\theta}_{\pi_b} \leftarrow \boldsymbol{\theta}_{\pi_t \pmod D}$

---

**Algorithm 3** Bernoulli Sampling

1: **Input:** $\{\boldsymbol{\theta}_{\pi_i}\}_{i=1}^D$, $t$, $\{p_{t,i}\}_{i=1}^D$
2: Sample $\boldsymbol{z} \sim \texttt{Bernoulli}(\{p\})$
3: Select $\boldsymbol{\theta}_{\pi_b} \leftarrow \boldsymbol{\theta}_{\pi_{\boldsymbol{z}==1}}$

---

**Algorithm 4** GSD Selection

1: **Input:** $\{\boldsymbol{\theta}_{\pi_i}\}_{i=1}^D$, current step $t$, $\{\nabla\mathcal{L}_i\}_{i=1}^d$, $\{\boldsymbol{\Sigma}_{\pi_i}\}_{i=1}^D$
2: $S_j \leftarrow 0$ for $j = \pi_1, \ldots, \pi_D$
3: **for** $j = \pi_1, \ldots, \pi_D$ **do**
4:     $S_j \leftarrow \sum_{i \in \text{block} j} \frac{|\nabla\mathcal{L}_i|^2}{(\boldsymbol{\Sigma}_{\pi_j})_i}$
5: **end for**
6: $k^* \leftarrow \arg\max_j S_j$
7: Select $\boldsymbol{\theta}_{\pi_b} \leftarrow \boldsymbol{\theta}_{k^*}$

---

**Algorithm 5** Recap of Bandit Method

1: **Input:** bandit consts, $\{p_{t,i}\}_{i=1}^d$
2: **for** active block $d$ **do**
3:     $\tilde{l}_{t,i} \leftarrow -\frac{||\nabla\mathcal{L}_i||^2}{(p_{t,i})^2} + \frac{L_{\text{bound}}^2}{p_{\min}^2}$
4:     $w_{t,i} \leftarrow p_{t,i} \exp\left(-\alpha_p \tilde{l}_{t,i}/p_{t,i}\right)$
5: **end for**
6: $p_{t+1} \leftarrow \arg\min_{q \in \mathcal{P}} D_{\text{KL}}(q \| w_t)$

---

**Remark 1.** The optimization objective is to minimize the loss function $\mathcal{L}(\boldsymbol{\theta})$. During each iteration, after selecting the active blocks for updates, zeroth-order optimization with a diagonal Hessian preconditioner is performed for the chosen layers. The diagonal Hessian estimate will be reinitialized for a newly selected block, and updates for that block will occur over several subsequent iterations. The algorithm applies in-place perturbations to the parameters in three steps with the perturbation scales correspondingly to $\mu_i = 0, +\mu, -2\mu$, sampling a normally distributed random vector $\boldsymbol{z}$ to perturb the selected block $\boldsymbol{\theta}_{\pi_b}$. With the perturbations, the loss function $\mathcal{L}(\boldsymbol{\theta})$ is computed to estimate the gradient and Hessian information. The gradient for the selected block is then projected using the Hessian, and the weights of the active block will update accordingly. Afterwards, the diagonal Hessian is updated based on the difference in the computed losses from the perturbed parameters.

**Remark 2.** The proposed algorithm efficiently combines BCD with a zeroth-order Newton method by updating only a subset of model layers per iteration. This approach reduces memory usage by eliminating backpropagation and utilizing blockwise gradient updates. It maintains convergence

speed through the use of diagonal Hessian approximations. The consistent use of random vectors and selective parameter perturbation further enhances the method's memory efficiency.

### 4.2 CONVERGENCE ANALYSIS

Our proposed method's convergence is analyzed based on its formulation as a randomized BCD algorithm. While this work emphasizes practical implementation and memory efficiency, we concisely summarize its convergence properties. We adopt standard assumptions for BCD, such as block Lipschitz continuity. The parameters of a selected block $i$ are updated using $\boldsymbol{\theta}_{t+1,[i]} = \boldsymbol{\theta}_{t,[i]} - \eta_t \tilde{\boldsymbol{g}}_{t,[i]}$. The gradient estimate is obtained via the proposed ZO-BCD-Newton approach:

$$\tilde{\boldsymbol{g}}_{t,[i]} = \frac{Z_{t,i}}{p_i} \cdot \frac{\mathcal{L}\left(\boldsymbol{\theta}_t + \mu \sum_{d=1}^{D} \boldsymbol{\Sigma}_{t,[d]}^{1/2} \boldsymbol{z}_{[d]}\right) - \mathcal{L}\left(\boldsymbol{\theta}_t - \mu \sum_{d=1}^{D} \boldsymbol{\Sigma}_{t,[d]}^{1/2} \boldsymbol{z}_{[d]}\right)}{2\mu} \cdot \boldsymbol{\Sigma}_{t,[i]}^{1/2} \boldsymbol{z}_{[i]}, \quad (5)$$

where $Z_{t,i} \sim \text{Bernoulli}(p_i)$ is the sampling indicator for block $i$, $\mu$ is the smoothing radius, and $\boldsymbol{\Sigma}_{t,[i]}$ is the block-diagonal Hessian approximation. With an appropriate step size and effective block sampling probabilities, our method achieves:

$$\mathbb{E}\left[\frac{1}{T}\sum_{t=1}^{T}\|\nabla\mathcal{L}(\boldsymbol{\theta}_t)\|_{\boldsymbol{\Sigma}_t}^2\right] \leq \mathcal{O}\left(\frac{\mathcal{L}(\boldsymbol{\theta}_1) - \mathcal{L}^*}{T}\right) + \mathcal{O}\left(\frac{1}{\sqrt{T}}\right), \quad (6)$$

where $\mathcal{L}^*$ denotes the minimum of $\mathcal{L}(\boldsymbol{\theta})$, and $\|\cdot\|_{\boldsymbol{\Sigma}_t}^2$ is the $\boldsymbol{\Sigma}_t$-preconditioned squared norm. A detailed derivation is provided in the Appendix. For foundational theoretical details, we refer readers to Liu & Mozafari (2022) and Zhao et al. (2024b).

## 5 EXPERIMENTS

In this section, we build on the experimental settings of MeZO (Malladi et al., 2023) and HiZOO (Zhao et al., 2024b) to evaluate our proposed method in terms of memory consumption, runtime, and convergence. Our experimental code builds on their open-source repositories, integrating the method. To facilitate implementation and reduce resource requirements, we follow their approach of focusing on performance across several GLUE and SuperGLUE tasks. All experiments are conducted on RTX 4090 (24GB) or RTX A6000 (48GB) GPUs. Specific details regarding the hyperparameter grids and implementations are provided in the appendix.

### 5.1 EXPERIMENTS ON OPT-1.3B

**Settings.** First, we fine-tuned OPT-1.3B on a single RTX 4090 GPU for selected GLUE/SuperGLUE tasks. First-order (FO) baselines include SGD, BCD-SGD, LoRA, and a vanilla AdamW-HF-Based BAdam version we implemented. Zeroth-order (ZO) methods compared are MeZO, HiZOO, and our proposed FOCUS. Batch sizes are 8 for ZO methods and 2 for FO methods to manage memory.

Table 2: Experiments on OPT-1.3B on SST-2 dataset.

| | Method | | Accuracy | Runtime | | Average Memory Cost | |
|---|---|---|---|---|---|---|---|
| First-Order | Forward+Backward | SGD | 94.3 | 4min 05s | | 22.7 GB | high memory demand |
| | | BCD-SGD | 92.4 | 3min 09s | | 20.5 GB | high memory demand |
| | | LoRA | 92.0 | 0min 55s | | 10.6 GB | not full fine-tuning |
| | | BAdam* | 93.7 | - | | 10.5 GB | bs=8, *ours implementation |
| Zeroth-Order | 2×Forward | MeZO | 91.7 | 54min 55s | baseline | 4.4 GB | baseline |
| | 3×Forward | HiZOO | 91.7 | 99min 44s | + 81.61% | 7.5 GB | + 72.25% |
| | 3×Forward | **ours** | **91.9** | **51min 38s** | **- 6.98%** | **4.1 GB** | comparable with MeZO |

**Memory Efficiency.** For memory efficiency, FOCUS significantly reduces the memory overhead of incorporating Hessian information while maintaining accuracy. As shown in Tables 1 and 2, our report on average GPU memory usage during experiments demonstrates that FOCUS has a comparable memory cost to MeZO, while offering substantial savings in memory consumption compared to HiZOO and FO methods such as SGD, BCD-SGD, and LoRA (`rank`=8). This notable improvement

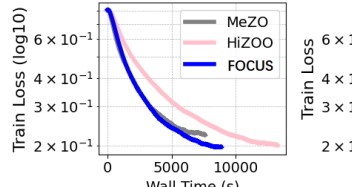
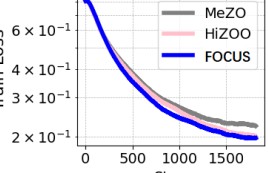

Figure 1: Average GPU memory consumption using different methods.

Figure 2: Convergence curves of MeZO, HiZOO, and proposed FOCUS on SST-2.

ensures the practical adoption of the proposed method on low-end devices, where memory is a primary bottleneck for training, which is also the original reason why the forward-only approach was developed to save memory down to inference-level requirements. This makes our method a suitable solution for low-memory training environments. In contrast, HiZOO incurs a significant 72% higher memory cost than MeZO, indicating an impractical convergence-memory tradeoff in memory-limited scenarios. Additionally, FO methods consume even more memory due to the overhead introduced by backpropagation. For instance, BCD-SGD still requires nearly complete fine-tuning memory to store activations and gradients for backpropagation. Consequently, their substantial memory demands render them impractical in low-end environments, making faster convergence irrelevant. This further highlights the advantages and rationale of our approach.

Table 3: Experiments on OPT-1.3B across different datasets.

| Task | Task Type | SST-2 | RTE | CB | BoolQ | WSC | WIC | SQuAD generation | Average |
|---|---|---|---|---|---|---|---|---|---|
| | | —————————-classification—————————- | | | | | | generation | |
| First-Order | SGD | 94.3 | 68.6 | 71.4 | 70.0 | 63.5 | 61.4 | 81.6 | 73.0 |
| | BCD | 92.4 | 69.7 | 69.6 | 63.2 | 63.5 | 61.6 | 78.8 | 71.3 |
| | LoRA | 92.4 | 66.4 | 69.6 | 66.8 | 63.5 | 58.5 | 80.5 | 71.1 |
| Zeroth-Order | MeZO | 91.7 | 64.3 | 69.6 | 65.5 | 63.5 | 57.7 | 77.9 | 70.0 |
| | HiZOO | 91.7 | 64.6 | 71.4 | 65.5 | 63.5 | 58.5 | 78.7 | 70.6 |
| | **ours** | **91.9** | **65.3** | **69.6** | 65.2 | **63.5** | **57.7** | **77.9** | **70.2** |

**Convergence Study.** Regarding convergence rate, we present the convergence curve relative to wall-clock time or steps for training on the SST-2 dataset, as illustrated in Figure 2. The results show that while HiZOO converges more effectively than MeZO for 20,000 steps, it requires nearly double the completion time. Conversely, our proposed method achieves better convergence than MeZO and matches the performance of HiZOO while maintaining MeZO's time efficiency. This speedup is attributed to applying the BCD strategy, which activates only a subset of layers, thereby reducing computational demands. In our experiment, the subset consists of two layers per iteration. As a result, our method benefits from both zeroth-order and Newton methods, thanks to the use of BCD. Furthermore, the results presented in Table 3 and visualized in the appendix demonstrate that our method achieves comparable accuracy to baseline methods across benchmarks. While FO methods yield superior results, their memory consumption is several times higher than that of ZO methods, making them impractical for low-end environments. In contrast, our method improves memory efficiency while enhancing convergence, outperforming the MeZO baseline and offering a practical, efficient solution for low-end settings. These findings position FOCUS as a memory-efficient optimizer and an effective alternative to MeZO.

## 5.2 PERFORMANCE ON LARGER MODELS

To evaluate FOCUS on larger models and further analyze memory demands, we conducted experiments on LLaMA-2-7B, LLaMA-3-8B, and OPT-30B. The comprehensive results, detailing performance metrics under various settings, are presented in Table 4 and Table 5.

**Full precision LLaMA-2-7B Performance.** Fine-tuning LLaMA-2-7B in full precision on SST-2 with batch size 1, FOCUS achieved 90.6% accuracy using 32GB of memory (on a single RTX A6000 GPU). As shown in Table 4, this surpassed MeZO (85.2%, 31GB) in accuracy with comparable memory. LoRA (rank=8) achieved higher accuracy (94.8%) but also used more memory (41GB).

Table 4: Full precision ZO vs. FO Fine-tuning LLaMA-2-7B on SST-2 (RTX A6000 48GB).

| Zeroth-Order Method | Accuracy | Average Memory | | First-Order Method | Accuracy | Average Memory | |
|---|---|---|---|---|---|---|---|
| MeZO | 85.2 | **31GB** | baseline | LoRA | 94.8 | 41GB | +32.3% |
| FOCUS (ours) | **90.6** | 32GB | + 3.23% | OOM for Full SGD, BCD, and HiZOO. | | | |

Full FO methods (SGD, BCD) and HiZOO encountered out-of-memory (OOM) errors. Note that the limited batch size impacts absolute accuracy for all methods.

**First-Order Memory Footprint vs. Zeroth-Order.** We further conduct experiments fine-tuning LLaMA-3-8B (FP32/BF16, RTX A6000) to prove the substantial memory footprint of SOTA memory-efficient FO methods compared to ZO approaches, as illustrated in Table 5. Even memory-efficient FO baselines like GaLore-AdamW (Zhao et al., 2024a) (BF16, **bs**=8) and LOMO (Lv et al., 2023b) (**bs**=4) consume significantly more memory than MeZO (**bs**=128), while the vanilla BAdam (Luo et al., 2024) has high I/O costs between CPU and GPU memory in our test environment (consumer-grade motherboard and RAM), leading to system instability. This difference, primarily due to FO methods storing states for backpropagation, highlights challenges in low-resource environments.

Table 5: Results on LLaMA-7B/8B and OPT-30B.

**LLaMA-3-8B** (SST-2, RTX A6000)

| Method | Type | Memory | batchsize | Settings |
|---|---|---|---|---|
| GaLore | FO | OOM | 1 | FP32 |
| GaLore | FO | 42.1GB | 8 | BF16; Max **bs** |
| LOMO | FO | 39.9GB | 4 | Max **bs** |
| MeZO | ZO | 35.6GB | 128 | Max **bs** |

**OPT-30B** (SST-2, A100 GPUs

| Method | Type | Accuracy | Time | Settings |
|---|---|---|---|---|
| MeZO | ZO | 90.6% | 13.7h | 2×A100 |
| HiZOO | ZO | 90.3% | 20.8h | 2×A100 |
| **FOCUS** | ZO | **92.9%** | **7.5h** | 2×A100 |
| **FOCUS** | ZO | **93.6%** | **9.9h** | 8×A100 |

**Scalability and Efficiency.** In addition, we demonstrate FOCUS's scalability fine-tuning OPT-30B on SST-2, the results are in the second subtable of Table 5. On 2×A100 80GB nodes (total batch size 32), FOCUS achieved 92.89% accuracy in just 7.5 hours. This significantly outperformed MeZO (90.6% accuracy, 13.7 hours) and HiZOO (90.3% accuracy, 20.8 hours) (Zhao et al., 2024b), marking substantial speedups alongside superior accuracy. Furthermore, on a larger 8× A100 80GB setup (total batch size 128), FOCUS attained 93.6% accuracy in 9.9 hours. This performance rivals the accuracy reported by HiZOO for the much larger OPT-66B model (93.6%), underscoring FOCUS's strong efficiency and generalizability. These results collectively demonstrate FOCUS's strength as a memory-efficient, scalable, and high-performing optimizer.

## 6 CONCLUSION

This paper proposes a novel memory-efficient zeroth-order Newton method for fine-tuning large language models. It integrates a novel block coordinate descent (BCD) scheme with Gauss-Southwell-based bandit sampling to build a diagonal Hessian-preconditioned zeroth-order optimizer. Our approach mitigates the substantial memory overhead commonly associated with second-order methods by employing selective blockwise updates. By combining the BCD technique with the Hessian pre-conditioner, we achieve significant reductions in memory consumption while maintaining competitive accuracy and convergence speed. Our extensive experiments on OPT-1.3B/30B and LLaMA-2-7B demonstrate that our method can reduce memory usage by up to 40% compared to existing second-order optimizers while maintaining baseline accuracy across various downstream tasks. Furthermore, our approach exhibits faster wall-clock convergence than conventional zeroth-order methods, making it a practical and scalable solution for fine-tuning large models on resource-constrained devices. Future work will aim to extend this methodology to larger models and more complex tasks, refining the block selection strategies to further enhance efficiency and performance. Our method provides a promising direction for memory-efficient fine-tuning of LLMs, offering practical advantages, particularly in memory-limited environments.

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

## A   APPENDIX

### A.1   ETHICS STATEMENT

All authors have read and adhered to the ICLR Code of Ethics. This work focuses on improving the memory efficiency of fine-tuning large language models, with potential positive societal impacts such as making AI more accessible on resource-constrained devices. The research does not involve human subjects, sensitive data, or unethical applications.

### A.2   REPRODUCIBILITY STATEMENT

To support reproducibility, we have included detailed descriptions of our method (Section 4), hyperparameter settings, and experimental setups. Pseudocode for the proposed algorithm is provided. Additional results, convergence analysis, and implementation details are available in the appendix.

### A.3   USE OF LARGE LANGUAGE MODELS

LLMs are used solely for polishing the language and improving the readability of the manuscript. All scientific contributions are entirely our own.

## B   PIPELINE ILLUSTRATION AND IMPLEMENTATION DETAILS

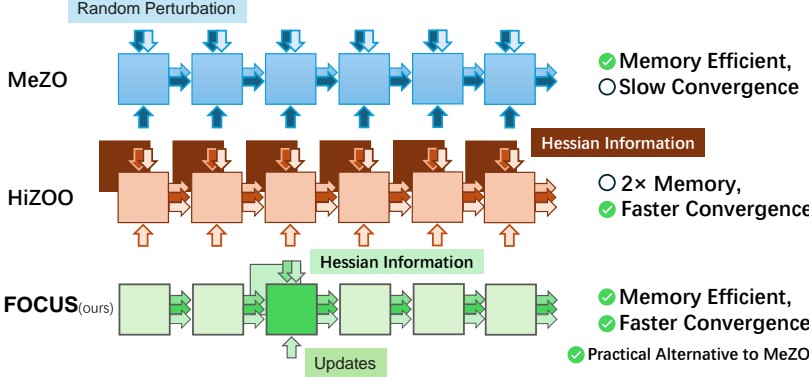

Figure 3: Illustration of MeZO, HiZOO, and our proposed FOCUS training pipeline. Our method leverages block coordinate descent to optimize subsets of model layers (blocks) using Hessian-informed zeroth-order updates, thereby reducing computational and memory costs.

### B.1   IMPLEMENTATION OVERVIEW

The implementation of our proposed FOCUS optimizer is designed to enhance zeroth-order optimization by incorporating selective layer-wise (block-wise) updates, informed by diagonal Hessian approximations. This approach builds upon concepts from MeZO (Malladi et al., 2023) for memory-efficient forward passes and HiZOO (Zhao et al., 2024b) for incorporating second-order information, but with a crucial block coordinate descent (BCD) strategy to manage memory and computation.

In our framework, at each iteration, a subset of model layers (blocks) is selected for updates. For these active blocks, we maintain and update an estimate of the diagonal Hessian. If a block is selected for the first time or after a certain period of inactivity, its diagonal Hessian estimate $\Sigma_{\pi_b}$ is initialized (e.g., to an identity matrix of the block's dimension $|\theta_{\pi_b}|$, as shown in Algorithm 1 of the main paper).

The core update mechanism for an active block $\theta_{\pi_b}$ involves:

1. **Perturbation**: Similar to MeZO and HiZOO, we apply noise-based perturbations to the parameters of the selected active block(s). Three forward passes are performed: one at the

current parameters $\theta_{\pi_b}$, one at $\theta_{\pi_b} + \mu \boldsymbol{\Sigma}_t^{1/2} \boldsymbol{z}$, and one at $\theta_{\pi_b} - \mu \boldsymbol{\Sigma}_t^{1/2} \boldsymbol{z}$. The random vector $\boldsymbol{z}$ is generated using a seed $s$ for reproducibility and memory saving, and $\boldsymbol{\Sigma}_t^{1/2}$ is the square root of the current diagonal Hessian estimate for the active block.

2. **Gradient Estimation**: The projected gradient for the active block is estimated using the losses from the perturbed forward passes: projected_grad $\leftarrow (l_+ - l_-)\Sigma_t^{1/2}\boldsymbol{z}/(2\mu)$.

3. **Hessian Update**: The diagonal Hessian estimate for the active block is updated using the formula $\hat{\Sigma}_t \leftarrow \frac{\Delta l}{2\mu^2}\Sigma_{t-1}^{-1/2}\boldsymbol{z}\boldsymbol{z}^\top\Sigma_{t-1}^{-1/2}$ (simplified following Zhao et al. (2024b)) and an exponential moving average (EMA): $\Sigma_t^{-1} \leftarrow (1 - \alpha_t)\Sigma_{t-1}^{-1} + \alpha_t|\hat{\Sigma}_t|$.

4. **Parameter Update**: The parameters of the active block are updated using the estimated projected gradient: $\theta_i \leftarrow \theta_i - \eta_t \cdot$ projected_grad$_i \cdot z_i$ for $\theta_i \in \theta_{\pi_b}$.

The selection of active blocks $\theta_{\pi_b}$ can follow various BCD rules. The main paper discusses Gauss-Seidel (cyclic selection), randomized selection, and score-based selection like the Gauss-Southwell-Diagonal (GSD) rule. The GSD rule, which prioritizes blocks based on their saliency scores $S_j \leftarrow \sum_{i \in \text{block}_j} \frac{|\nabla\mathcal{L}_i|^2}{(\Sigma_{\pi_j})_i}$ (where $\nabla\mathcal{L}_i$ is estimated via ZO methods for the block), can be computationally intensive if scores for all blocks need to be evaluated frequently. To mitigate this, we employ a bandit-based method for efficient probabilistic layer selection, detailed next.

We utilize `torch.clamp` for intermediate results involving the Hessian to ensure numerical stability and meet precision requirements, which is particularly important for second-order methods. Memory management, such as periodically calling `torch.cuda.empty_cache()`, is employed to maintain efficiency in memory-constrained environments, although the primary memory saving comes from the BCD approach itself by not storing Hessians for all layers.

## B.2 BANDIT METHOD FOR GSD-BASED LAYER SELECTION

To efficiently implement the Gauss-Southwell-Diagonal rule for selecting active blocks without computational overhead for all $D$ blocks at every iteration, we adapt a bandit method inspired by Liu & Mozafari (2022), originally from `auer2002finite`. This method updates a probability distribution $p_t = [p_{t,1}, \ldots, p_{t,D}]$ over the $D$ blocks of the model. At each iteration $t$, we sample a small set of active blocks $S_t$ according to $p_t$ (or select the top-k based on $p_t$ if only one or a few blocks are updated). The Gauss-Southwell induced scores are then computed only for these active blocks.

Let $S_{j,t}$ be the score for block $j$ if it's in the active set $S_t$ at iteration $t$. The bandit algorithm updates the probability distribution $p_t$ based on these observed scores. The objective is to increase the probability of selecting blocks that have higher scores, thus focusing computational resources on more important blocks.

The update mechanism is detailed in Algorithm 6. We define a set $\mathcal{P} = \{q \in \mathbb{R}^D : \sum_{d=1}^D q_d = k, q_d \geq p_{\min}, \forall 1 \leq d \leq D\}$, where $p_{\min}$ is a minimum probability to ensure exploration. For simplicity, we assume $k = 1$ here (one block selected per step).

The "loss" $\tilde{l}_{t,d}$ for the bandit algorithm is inversely related to the Gauss-Southwell rule: a higher score means a lower loss (i.e., a better arm). We use an upper-bound Score$_{t,d}$ to denote the score of block $d$ at iteration $t$, and $S_{\text{upper}}^2$ ensures non-negative weights.

The KL divergence projection step ensures $p_{t+1}$ remains a valid probability distribution. This bandit-based selection helps to dynamically focus on important layers according to the GSD rule without the prohibitive cost of evaluating all layers at each step.

## C HYPERPARAMETER SEARCH

Here, we present the detailed hyperparameter grids used in our experiments, as shown in Table 6. Empirically, we found that the optimal learning rate for FOCUS is often an order of magnitude higher than that for MeZO. Some outlier values in the results may stem from insufficient parameter search or

---

**Algorithm 6** Bandit Method for Updating Block Selection Distribution based on GSD Scores.

---

1: **Algorithm:** UPDATEBANDITDISTRIBUTION($p_t$, $S_t$, $\{\text{Score}_{t,d}\}_{d \in S_t}$)
2: **Input:** $p_t$: current probability distribution over $D$ blocks.
      $S_t$: set of active block(s) for which GSD scores were computed at iteration $t$.
      $\{\text{Score}_{t,d}\}_{d \in S_t}$: GSD scores for blocks in $S_t$.
      $\alpha_p$: learning rate for bandit updates.
      $p_{\min}$: minimum probability for any block.
      $S_{\text{upper}}$: an estimated upper bound for GSD scores (for normalization).
3: **for** $d = 1$ **to** $D$ **do**
4:     **if** $d \in S_t$ **then**
5:       $\tilde{l}_{t,d} = -\frac{(\text{Score}_{t,d})^2}{(p_{t,d})^2} + \frac{S_{\text{upper}}^2}{p_{\min}^2}$           ▷ Higher GSD score = better block = lower loss
6:     **else**
7:       $\tilde{l}_{t,d} = \frac{S_{\text{upper}}^2}{p_{\min}^2}$           ▷ Default loss for non-evaluated blocks, or 0 if no penalty
8:     **end if**
9:     $w_{t,d} = p_{t,d} \exp\left(-\alpha_p \tilde{l}_{t,d}/p_{t,d}\right)$       ▷ Avoid division by $p_{t,d}$ if not used in loss
10: **end for**
11: $p_{t+1} = \arg\min_{q \in \mathcal{P}} D_{\text{KL}}(q\|w_t)$       ▷ Project $w_t$ back to the probability simplex $\mathcal{P}$
12: **return** $p_{t+1}$

---

incomplete convergence, likely caused by limited training steps and small batch sizes due to hardware memory constraints.

| Model | Method | Hyperparameters | Values |
|---|---|---|---|
| General Settings in Common | | LR schedule
Steps
LoRA rank | Linear decay
20000
8 |
| OPT-1.3B | FO Baselines | Batch size
Learning rate
$\mu$
Weight Decay | $\{1, 2\}$
$\{1, 3, 5, 7\} \times \{1e{-}6, 1e{-}7\}$
$1e{-}3$
0 |
| OPT-1.3B | ZO | Batch size
Learning rate
$\mu$
Weight Decay
$\alpha_t$
BCD-Update Interval
layers per update | $\{1, 2, 8\}$
$\{1, 3, 5, 7\} \times \{1e{-}5, 1e{-}6\}$
$1e{-}3$
0
$1e{-}9$ (HiZOO), $1e{-}5$ (FOCUS)
$\{5, 10\}$ steps
$\{1, 2\}$ |
| LLaMA-2-7B | All Methods | Batch size
Learning rate
$\mu$
Weight Decay | $\{1\}$
$\{3\} \times \{1e{-}6, 1e{-}7\}$
$1e{-}3$
0 |

Table 6: The hyperparameter grids used for OPT-1.3B and LLaMA-2-7B experiments. Note: $\mu$ is primarily for zeroth-order methods. Learning rates for first-order and zeroth-order methods often differ.

# D  ADDITIONAL VISUALIZATION RESULTS

Here, we present the bar chart illustrating the test results of fine-tuning OPT-1.3B with different methods across various GLUE and SuperGLUE benchmarks, as shown in Figure 4. This complements the results presented in the main paper.

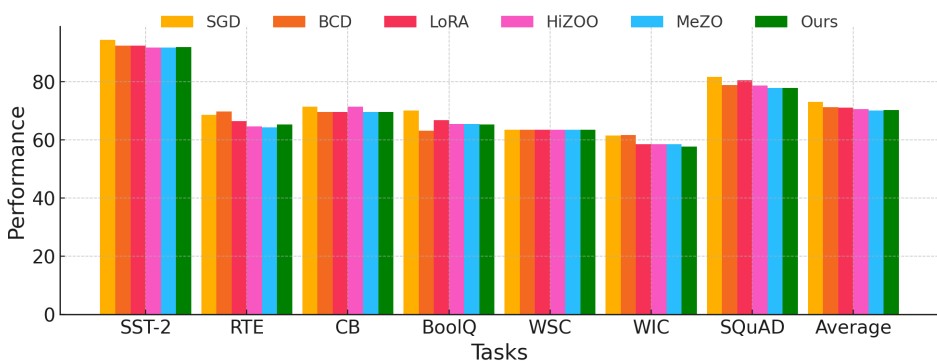

Figure 4: Bar chart illustrating the performance of OPT-1.3B fine-tuned with different methods (SGD, BCD-SGD, LoRA, MeZO, HiZOO, and FOCUS (ours)) across various downstream tasks.

## E   CONVERGENCE ANALYSIS

We provide a convergence analysis for the randomized GS-BCD version of our algorithm, considering necessary assumptions from Liu & Mozafari (2022) and Zhao et al. (2024b):

- Block Lipschitz: The objective function $\mathcal{L}(\boldsymbol{\theta})$ is $L_i$-smooth with respect to each block $\boldsymbol{\theta}_i$. Let $L_\infty = \max_i L_i$.

- The stochastic zeroth-order gradient estimate has bounded variance.

- Bounded Hessian Approximation: Each entry of the diagonal Hessian approximation $\Sigma_t$ for an active block lies in the range $[\beta_\ell, \beta_u]$ with $0 < \beta_\ell \leq \beta_u$.

Suppose the model has $D$ blocks. We denote $\theta_{t,[i]} = [0, \ldots, \theta_{t,i}, \ldots, 0]$ as the parameters for the $i$-th block embedded in the full parameter space at iteration $t$. Similarly, $g_{t,[i]} = [0, \ldots, g_{t,i}, \ldots, 0]$ is the gradient for the $i$-th block. This bracket notation $[\,]$ also applies to the diagonal Hessian approximation $\Sigma_t$ and the random perturbation vector $\boldsymbol{z}_t$. Each block $i$ has a probability $p_{t,i}$ of being selected at iteration $t$. Let $Z_{t,i} \sim \text{Bernoulli}(p_{t,i})$ be the sampling indicator for block $i$. The gradient estimate is:

$$\tilde{g}_{t,[i]} = \frac{Z_{t,i}}{p_i} \cdot \frac{\mathcal{L}\left(\theta_t + \mu \sum_{d=1}^{D} \Sigma_{t,[d]}^{1/2} u_{[d]}\right) - \mathcal{L}\left(\theta_t - \mu \sum_{d=1}^{D} \Sigma_{t,[d]}^{1/2} u_{[d]}\right)}{2\mu} \cdot \Sigma_{t,[i]}^{1/2} u_{[i]}. \qquad (7)$$

From Taylor expansion:

$$\Delta\mathcal{L} = \mathcal{L}\left(\theta_t + \mu \sum_{d=1}^{D} \Sigma_{t,[d]}^{1/2} u_{[d]}\right) - \mathcal{L}\left(\theta_t - \mu \sum_{d=1}^{D} \Sigma_{t,[d]}^{1/2} u_{[d]}\right) = 2\mu \nabla^\top \mathcal{L}(\theta_t) \sum_{d=1}^{D} \Sigma_{t,[d]}^{1/2} u_{[d]} + \mathcal{O}(\mu^3). \tag{8}$$

The expectation yields:

$$\tilde{g}_{t,[i]} = \frac{Z_{t,i}}{p_i} \left( \sum_{d=1}^{D} \Sigma_{t,[i]}^{1/2} u_{[i]} u_{[d]}^\top \Sigma_{t,[d]}^{1/2} \nabla\mathcal{L}(\theta_t) + \mathcal{O}(\mu^2) \right), \tag{9}$$

$$\mathbb{E}[\tilde{g}_{t,[i]}] = \Sigma_{t,[i]} \nabla_i \mathcal{L}(\theta_t) + \mathcal{O}(\mu). \tag{10}$$

The update rule is given by:

$$\theta_{t+1,[i]} = \theta_{t,[i]} - \eta_t \cdot \tilde{g}_{t,[i]}. \tag{11}$$

Let $\tilde{g}_t = \sum_{i=1}^{D} \tilde{g}_{t,[i]}$. Under the block Lipschitz assumption with constant $L_\infty$:

$$\mathcal{L}(\theta_{t+1}) - \mathcal{L}(\theta_t) \leq \sum_{i=1}^{D} \left\langle \nabla_i \mathcal{L}(\theta_t), \theta_{t+1,[i]} - \theta_{t,[i]} \right\rangle + \frac{L_\infty}{2} \|\theta_{t+1} - \theta_t\|^2$$

$$= -\eta_t \sum_{i=1}^{D} \left\langle \nabla_i \mathcal{L}(\theta_t), \tilde{g}_{t,[i]} \right\rangle + \frac{L_\infty \eta_t^2}{2} \|\tilde{g}_t\|^2. \tag{12}$$

Take expectation and according to the proof in Zhao et al. (2024b):

$$\mathbb{E}[\mathcal{L}(\theta_{t+1})] - \mathbb{E}[\mathcal{L}(\theta_t)] \leq -\eta_t \|\nabla_i \mathcal{L}(\theta_t)\|_{\Sigma_t}^2 + \eta_t \mathcal{O}(\mu\|\nabla\mathcal{L}(\theta_t)\|) + \frac{L_\infty \eta_t^2}{2}\mathbb{E}[\|\tilde{g}_t\|^2]$$

$$\leq -\frac{\eta_t}{2}\|\nabla\mathcal{L}(\theta_t)\|_{\Sigma_t}^2 + \frac{L_\infty \eta_t^2}{2}\mathbb{E}[\|\tilde{g}_t\|^2]. \tag{13}$$

Summing equation 13 over $t = 1, \ldots, T$:

$$\sum_{t=1}^{T} \frac{\eta_t}{2}\|\nabla\mathcal{L}(\theta_t)\|_{\Sigma_t}^2 \leq \mathbb{E}[\mathcal{L}(\theta_1)] - \mathbb{E}[\mathcal{L}(\theta^*)] + \frac{L_\infty \eta_t^2}{2} \sum_{t=1}^{T} \sum_{i=1}^{D} \frac{\|\tilde{g}_{t,[i]}\|^2}{p_i}. \tag{14}$$

The optimal probabilities $p_{t,i}$ minimizing the second term are:

$$p_{t,i}^* = \frac{\|g_{t,[i]}\|}{\sum_{j=1}^{D} \|g_{t,[j]}\|}. \tag{15}$$

Liu & Mozafari (2022) solves this via bandit optimization and achieves an $\mathcal{O}\left(\frac{1}{\sqrt{T}}\right)$ convergence component. This leads to the convergence rate stated in the main paper:

$$\mathbb{E}\left[\frac{1}{T}\sum_{t=1}^{T} \|\nabla\mathcal{L}(\theta_t)\|_{\Sigma_t}^2\right] \leq \mathcal{O}\left(\frac{\mathcal{L}(\theta_1) - \mathcal{L}^*}{T}\right) + \mathcal{O}\left(\frac{1}{\sqrt{T}}\right). \tag{16}$$

