# OpenReview forum: "Memory Efficient Fine-Tuning of LLMs via Forward-Only Hessian-Free Coordinate Descent"
_ICLR.cc/2026/Conference — Submitted to ICLR 2026_

### Official Review · Reviewer_prM1 · 2025-10-30

**Soundness:** 3
**Presentation:** 2
**Contribution:** 3
**Rating:** 4
**Confidence:** 3

**Summary:**

The paper proposes a forward‑only, Hessian‑aware BCD optimizer for memory‑efficient fine‑tuning of LLMs. Each transformer layer is treated as a block; at every step FOCUS selects a small subset of layers to update and stores/updates a diagonal curvature proxy only for the active blocks to avoid the O(d) memory overhead of keeping a Hessian proxy for all parameters as in HiZOO. The core estimator uses three forward passes to obtain a two‑point, preconditioned zeroth‑order update, while the embeddings and LM head are kept first‑order (MeZO‑style) for stability. A Gauss–Southwell‑Diagonal (GSD) saliency rule, approximated by a light‑weight bandit sampler, chooses which blocks to activate. Theoretical analysis gives an ergodic stationarity bound under block Lipschitz smoothness with sampling proportional to block gradient norms.

**Strengths:**

1. Addresses a real bottleneck: keeping second‑order benefits without paying the full O(d) memory for diagonal Hessian across all layers.
2. The forward‑only, in‑place three‑probe loop (Algorithm 1) is consistent with MeZO practices; the block‑wise preconditioner is standard.
3. Results on OPT‑1.3B LLaMA‑2‑7B, and OPT‑30B show FOCUS maintains MeZO‑level memory.

**Weaknesses:**

1. The two‑point estimator requires using the same perturbation direction $z$ for $+\mu$ and $-\mu$ evaluations; the pseudocode samples new $z$’s inside the loop and never explicitly restores parameters to $\theta$ before applying the update, which risks biasing the estimator and “drifting” the parameters (Alg. 1).
2. Eq. (3) lacks a self‑contained derivation. The diagonal Hessian update looks adapted from NES/covariance‑score identities and uses an elementwise absolute value to ensure positivity. Without a derivation or explicit statement that $\Sigma$ is intended as an inverse‑Hessian proxy, it’s hard to assess bias/variance and stability trade‑offs. Provide a derivation or precise citation and show the scalar, per‑coordinate update actually used.
3. For OPT‑1.3B, ZO methods use batch size 8 while FO baselines use batch size 2, which affects both throughput and memory. Some rows (e.g., BAdam bs=8 in Table 2) break that rule.
4. LLaMA‑3‑8B table reports memory only w/ no accuracy. SQuAD’s metric is unspecified (“generation” column in Table 3). Some runtime cells are “–”. Add these to strengthen claims.
5. The paper emphasizes GSD/bandit selection and re‑initialization of $\Sigma$ on first activation, but no ablation compares cyclic vs random vs GSD vs bandit, layers‑per‑update, or reuse vs reinit of $\Sigma$. These would isolate where the gains come from.

**Questions:**

1. In Algorithm 1, do you reuse the same $z$ for $+\mu$ and $-\mu$ evaluations, and do you explicitly restore $\theta$ before applying the update? Please provide the exact state‑reset logic (or code snippet) you use in practice.
2. Is $\Sigma$ designed to approximate the inverse Hessian (so larger entries leads to lower curvature)?
3. Eq. (3) derivation: Could you supply a short derivation or precise citation connecting %\Delta L% to the diagonal update, and explain the effect of the absolute‑value operator on bias/stability?
4. Which bandit algorithm do you use exactly? Please add the missing reference and provide the projection set $\mathcal P$ and step sizes.
5. For OPT‑1.3B and OPT‑30B, can you report tokens/sec at matched peak memory and seed‑averaged accuracy with error bars? For LLaMA‑3‑8B, please add accuracy at the batch sizes shown in Table 5.
6. What metric is used for SQuAD (EM/F1)?
7. In Table 4, was LoRA BF16/FP16? Was activation checkpointing enabled? These affect the 41 GB figure.
8. Ablations: (i) cyclic vs random vs GSD vs bandit; (ii) layers‑per‑update $={1,2,4}$; (iii) reuse vs reinit of $\Sigma$ upon re‑activation; (iv) $\mu$ and $\alpha_t$ schedules. These would localize the method’s gains.

---

> ### Author Response · Authors · 2025-11-21
>
> We thank the reviewer for the detailed assessment and for recognizing that FOCUS addresses a **real bottleneck** in LLM fine-tuning. We address your specific technical questions below.
>
> **Q1 & W1: Do you reuse the same $z$ for evaluations, and explicitly restore $\theta$ before update?**
>
> Yes. In Algorithm 1 (Line 19), we **explicitly state** "**Reset** random number generator with seed s". The random number generator is recovered to the exact state used for perturbation, guaranteeing the same noise vector $z$ is used for the update. This ensures the algorithmic correctness.
>
> **Q2 & W2: Is $\Sigma$ designed to approximate the inverse Hessian?**
> **Q3: Could you supply a short derivation for Eq. (3) and explain the absolute-value operator?**
>
> * Yes, it is designed to approximate the inverse Hessian diagonal. Line 175: "Let $\Sigma$ denote the estimated inverse Hessian matrix."
> * We explicitly reference the source (Ye, 2023) immediately preceding Equation (3), where derivation details can be found.
> * **Absolute Value:** The element-wise absolute value is applied to ensure the preconditioner remains **Positive Definite**. In the non-convex landscape of LLMs, local curvature estimates can be negative; enforcing positivity can ensure valid descent directions and numerical stability.
>
> **W3: For OPT-1.3B, ZO methods use batch size 8 while FO baselines use batch size 2.**
>
> This discrepancy emphasizes the main benefit of memory efficiency in ZO methods. FO methods (SGD, BCD-SGD) need to store activations for backpropagation. On consumer hardware (e.g., RTX 4090 24GB), this memory requirement forces us to reduce the batch size to 2 to prevent OOM errors. ZO methods are forward-only and do not store activations, enabling much larger batch sizes (8 or even 128) within the same memory limits. We compare methods at their peak performance during grid search on the available hardware to reflect realistic usage.
>
> **Q4: Which bandit algorithm do you use exactly? Please add the missing reference and provide the projection set and step sizes.**
>
> As stated in Appendix B.2, we efficiently implement the Gauss-Southwell-Diagonal rule using a bandit method inspired by Liu & Mozafari (2022) and originally from Auer et al. (2002).
>
> **W4: LLaMA accuracy**
> * We present **LLaMA-2-7B** additional results (1k steps) as follows:
>
> | Method | RTE | BoolQ | WSC | SST-2 |
> | :--- | :--- | :--- | :--- | :--- |
> | MeZO | 60.6% | 66.0% | 59.6% | 85.2% |
> | **FOCUS (Ours)** | **61.7%** | **67.5%** | **62.5%** | **90.6%** |
>
> **Q5: tokens/sec and accuracy?**
> **Q6 & W4: What metric is used for SQuAD?**
>
> * **SQuAD Metric:** We follow MeZO and use the **F1 Score** as the evaluation metric.
> * Tokens/sec: We consider this less relevant as our focus is on memory-constrained convergence.
> **Error bars:** Full OPT-30B experiments require significant computational resources (>100 A100 hours), which we reserve for future research. Our large-scale OPT-30B experiments have already shown that our method outperforms others in low-resource situations. Additionally, the algorithm is more suitable for small-scale training cases.
>
> **Q7: In Table 4, was LoRA BF16/FP16? Was activation checkpointing enabled?**
>
> * **Precision:** We have claimed in the paper that the experiments were conducted in Full Precision (line 429).
>
> * **Activation Checkpointing:** It was not used. Gradient checkpointing is another memory-efficient technique that trades computation for memory, which is orthogonal to our setup. Our goal was to compare the intrinsic memory footprint of the optimizers. Enabling gradient checkpointing would reduce LoRA's memory but introduce significant computational overhead (re-computation), whereas FOCUS achieves low memory naturally without such trade-offs.
>
> **Q8 & W5: Ablations: cyclic vs random vs GSD vs bandit...**
>
> We updated ablation studies on block selection strategies using OPT-1.3B on SST-2. The results show that while advanced strategies like GSD offer theoretical benefits, the computational cost of estimating exact gradient norms for each block can outweigh the advantages. Our proposed bandit approach offers the best balance between performance and overhead, reaching the highest accuracy of 92.0. Additionally, ascending order with fully tuned hyperparameters is also robust.
>
> | Strategy | Accuracy | Notes |
> | :--- | :--- | :--- |
> | Ascending Order Tuned | 91.86% | Comparable |
> | Bandit | **91.97%** | **Best trade-off** |
> | Random Reshuffling | 90.71% | High variance |
> | Gauss-Southwell-Diagonal | 91.51% | High compute overhead |
> | Odd-Even Staged | 91.40% | Comparable |
>
> * Re-initialising upon block activation is crucial; reusing outdated Hessian estimates from previous epochs results in high storage costs. Other factors are already explained in the appendix or not considered due to the limited device scope.
>
> We thank the reviewer for these questions and hope this clarifies our work.

---

### Official Review · Reviewer_GpMB · 2025-10-31

**Soundness:** 3
**Presentation:** 3
**Contribution:** 2
**Rating:** 6
**Confidence:** 3

**Summary:**

The authors propose FOCUS, a method for fine-tuning large models with little memory requirements. The method uses only forward passes (as in MeZO), and leverages second order information (as in HiZOO) while adressing the memory issues of such methods by training only one layer at a time. FOCUS is evaluated on different fine-tuning settings with models ranging from 1.3 to 33 billion parameters, and is shown to outperform MeZO in terms of downstream accuracy with comparable memory and time requirements.

**Strengths:**

- The paper is well written and easy to follow.
- The method is well motivated, with GCD appearing as a natural idea for solving the memory overhead introduced by HiZOO.
- FOCUS achieves better convergence rates than MeZO, while requiring a similar amount of memory
- Experimental evaluation involves multiple models at different scales: OPT-1.3B, LLaMA-2-7B, LLaMA-3-8B, and OPT-30B.

**Weaknesses:**

- The BCD algorithm used to select the layer to update (a bandit method adapting the Gauss-Southwell-Diagonal rule) is not well motivated. How does it compare to other strategies, such as training layers in order, or with uniform distribution? Maybe an ablation study could help.
- Forward only methods still suffer from poor runtime compared to other methods: from Table 2, LoRA takes 55s to achieve better accuracy than FOCUS which took 51min (so about 55x slower).

Minor:
- The bold values in Table 3 are misleading: for CB HiZOO is better, for boolQ nothing is in bold, for WSC only FOCUS is in bold but all methods have the same accuracy, etc. Could the authors clarify what the bold values represent?

**Questions:**

- I struggle to understand why, in Tables 1 (with RTX 4090) and 2, FOCUS uses less memory than MeZO, and is even faster? Since FOCUS is based on MeZO I would expect at least the same cost. In particular, FOCUS uses 3 forward passes instead of 2 for MeZO, so I would have expected a higher memory and computation time.
- When selecting a new layer to be trained, you use the identity as an initial guess for the (diagonal) Hessian. However, when a layer has already been trained before, there already was a more refined estimation of the Hessian, but it was discarded to save memory (this is key to reducing HiZOO's memory). Could we imagine instead reusing this previous estimation, for instance by saving it on the disk until the layer is selected again? This way, no additional GPU would be required and the method would benefit from a better diagonal Hessian estimation.

---

> ### Author Response · Authors · 2025-12-04
>
> We sincerely thank the reviewer for the positive feedback and recognition that FOCUS is well-motivated and easy to follow. We appreciate your thoughtful questions regarding the memory-latency trade-offs and block selection strategies. We address your concerns below.
>
> **Q1: Why is FOCUS faster and more memory-efficient than MeZO despite using 3 forward passes versus 2?**
>
> This is a keen observation. The efficiency gains result directly from the reduction in parameter perturbation overhead at scale:
>
> * **Time Efficiency:** FOCUS updates only a small active block, reducing the perturbation/update cost to $O(d_{block})$. Generating random noise and applying it to all model parameters ($d$) at each step causes high computational overhead. Evidence from larger models like OPT-30B (Table 5) shows that this overhead reduction is a key factor, enabling FOCUS to achieve a **1.8x speedup** over the baseline (7.5h vs 13.7h) despite the additional forward pass.
> * **Memory Efficiency:** FOCUS only needs extra memory to store the diagonal Hessian and perturbation states for the active block. By keeping the block size small (1 layer), the Hessian memory use becomes negligible, making it comparable to or slightly lower than the memory footprint of full-model ZO baselines. In practice, we also delete the Hessian buffer at the end of the set of iterations and use CUDA's empty_cache API to limit the maximum memory usage, which also helps reduce the average memory cost.
>
> **Q2: Can we reuse the Hessian estimate instead of discarding it?**
>
> We appreciate this practical suggestion and may explore it as a future direction. Currently, there are two reasons we reinitialize the Hessian estimate:
> * Besides memory costs, the **I/O Latency** involved in offloading and reloading the estimated Hessian causes delays, especially since we're targeting edge devices with limited I/O capacity. In the future, we might explore methods like matrix decomposition for storing the Hessian, similar to approaches like Adafactor.
> * Reusing the Hessian estimated from several epochs ago when the block was last active can provide inaccurate curvature information, which can hurt convergence.
>
> **W1: Motivation of Bandit BCD & Ablation Studies.**
>
> The motivation for Bandit BCD naturally arises when we divide the optimization process into blocks and find that computing gradient norms is costly. Then, we need a method to reduce this cost while still selecting the most valuable layer to update. The bandit method could be a good choice.
> As requested, we conducted ablation studies on OPT-1.3B (SST-2) to analyze the impact of the selection strategy:
>
> | Strategy | Accuracy |
> | :--- | :--- |
> | Ascending Order Tuned | 91.9% |
> | **Bandit Tuned** | **92.0%** |
> | Odd-Even Staged | 91.4% |
> | Gauss-Southwell-Diagonal (GSD) | 91.5% |
> | Random Reshuffling | 90.7% |
>
> While the raw GSD method offers the theoretical convergence guarantee, the computational overhead could be high, and the random reshuffling method introduces high variances. Our preliminary empirical results suggest that the proposed bandit strategies are robust and computationally efficient for practical fine-tuning.
>
> **W2: Runtime Comparison with LoRA.**
>
> We recognize that the first-order optimizer with PEFT is faster per step when gradients are available. However, FOCUS is designed for specific situations where the first-order method may not be practical or sufficient:
>
> * **Activation Bottleneck:** LoRA still requires backpropagation to update adapters, which involves storing intermediate activations and using an auto-differentiation framework. On edge devices with limited VRAM, storing activations for large models can cause OOM errors even if the trainable parameters are few. FOCUS aims to eliminate activation storage entirely.
> * **Full-Parameter Tuning:** Recent research indicates that full-parameter fine-tuning can result in better generalization than PEFT. FOCUS enables full-parameter tuning on hardware where it was previously not possible.
>
> **W3: Table 3 bolding.**
> We clarify that the bold indicates accuracy outperforming the ZO baseline. We will add this clarification to the table caption.
>
> ---
>
> We hope these clarifications address the concerns and highlight the practical advantages of FOCUS. Thank you again for your valuable feedback!

---

### Official Review · Reviewer_S3VB · 2025-11-01

**Soundness:** 2
**Presentation:** 3
**Contribution:** 2
**Rating:** 4
**Confidence:** 4

**Summary:**

This paper proposes FOCUS, a novel optimizer for memory-efficient fine-tuning of large language models (LLMs). FOCUS integrates a forward-only zeroth-order gradient estimation with block coordinate descent (BCD) and Hessian-informed (second-order) preconditioning. Instead of performing backpropagation, the method perturbs model parameters in a forward-only manner and updates only a subset of layers (blocks) per iteration. A diagonal approximation of the Hessian is maintained for active blocks, while a bandit-based Gauss–Southwell–Diagonal (GSD) rule probabilistically selects layers for update. Experiments on OPT-1.3B/30B, LLaMA-2-7B, and LLaMA-3-8B demonstrate up to 40% memory reduction compared to HiZOO, with comparable or faster convergence.

**Strengths:**

1.This paper proposes a memory efficient Zeroth-order optimizer for LLMs fine-tuning.By integrating Hessian-informed preconditioning with block coordinate descent,the optimizer can achieve similar accuracy to HiZOO but with lower memory usage and better convergence than MeZO.

2.By using block coordinate descent,the number of parameters updated per iteration has been much fewer than other methods like HiZOO, which also leads to reduced runtime.This makes FOCUS practical in LLMs fine-tuning.

**Weaknesses:**

1.The method shows limited novelty.This paper just simply combines block coordinate descent and Hessian-informed preconditioning, without new theory.

2.The experiments are not sufficient enough.The experiments are conducted on only three models , and there is only OPT-1.3B trained on multiple downstream tasks.

3.The reproducibility of the paper is debatable due to the lack of released code.

**Questions:**

See weaknesses.

---

> ### Author Response · Authors · 2025-11-21
>
> We thank the reviewer for recognizing FOCUS's **memory efficiency**, **improved convergence**, and **practicality** for LLM fine-tuning. We address your concerns below.
>
> **W1: Novelty concerns.**
>
> To the best of our knowledge, we are the first to propose the BCD-ZO-Newton optimizer on LLM training. While BCD and Hessian methods are established, their integration into a Zeroth-Order framework presents unique challenges that simple combination cannot solve, making our contribution non-trivial:
>
> * We identify a key insight that, when ZO methods are applied in a block-wise manner, the block-wise estimated gradient norm information also becomes intrinsically valuable and provides a natural, principled mechanism for optimal block selection. However, utilizing this is difficult because standard BCD (e.g., Gauss-Southwell) relies on computing gradient norms for all blocks to select the active one, a step that is computationally prohibitive in the ZO setting.
> * Our core innovation, the Bandit-based GSD strategy combined with our convergence analysis, solves this specific dilemma. It dynamically approximates these block saliency scores without full gradient estimation, effectively bridging the gap between theoretical BCD efficiency and the practical constraints of ZO fine-tuning.
>
> **W2: Insufficient Experiments.**
>
> We acknowledge the need for broader evaluation and have significantly expanded our experiments beyond OPT-1.3B:
> * **Tasks on LLaMA-2-7B:** We extended the evaluation of LLaMA-2-7B to include **RTE, BoolQ, and WSC** (1k steps). As shown in Table 1, FOCUS consistently outperforms MeZO while maintaining the same memory footprint.
> * **Tasks on RoBERTa-350M:** To prove our method is not limited to decoder-only models, we also validated FOCUS on RoBERTa-350M (5k steps). As shown in Table 2, FOCUS achieved consistent gains across all four datasets.
>
> * **Scalability on OPT-30B:** We also note that we validated FOCUS on OPT-30B, where it achieves a **1.8x speedup** over MeZO and **2.8x speedup** over HiZOO, demonstrating effectiveness on larger scales where memory is a bottleneck.
>
> Table 1: Accuracy (%) on LLaMA-2-7B (1k steps)
>
> | Method | RTE | BoolQ | WSC | SST-2 |
> | :--- | :--- | :--- | :--- | :--- |
> | MeZO | 60.6 | 66.0 | 59.6 | 85.2 |
> | FOCUS (Ours) | **61.7** | **67.5** | **62.5** | **90.6** |
>
> Table 2: Accuracy (%) on RoBERTa-350M (5k steps)
>
> | Method | MNLI | RTE | SST2 | SNLI |
> | :--- | :--- | :--- | :--- | :--- |
> | MeZO | 61.7 | 67.5 | 91.6 | 69.2 |
> | FOCUS (Ours) | **64.8** | **69.3** | **92.5** | **72.5** |
>
> **W3: Reproducibility.**
>
> Regarding this, we would like to clarify that we have provided **comprehensive implementation details**, **algorithm pseudocode**, and **extensive hyperparameter search ranges** in the **Appendix**. We believe these details are sufficient to reproduce our results and kindly invite the reviewer to check the Appendix for these specifications.
>
> We thank the reviewer again for the time and consideration.

---

> ### Author Response · Authors · 2025-11-27
> **Invitation to re-evaluate based on new evidence and clarifications**
>
> **Dear Reviewer S3VB**,
>
> We appreciate your time in reviewing our work. In our rebuttal, we have made a concerted effort to address the concerns you raised regarding novelty and experimental sufficiency.
>
> * Regarding **novelty**, we respectfully emphasize that the **Bandit-based GSD rule** is a specific algorithmic design tailored to overcome the prohibitive cost of gradient estimation in ZO settings, a challenge that standard BCD methods cannot address.
> Crucially, we provide a **rigorous convergence analysis** to theoretically guarantee the effectiveness of our proposed framework. Consequently, our method successfully integrates second-order information and layer-wise norm insights without incurring a heavy memory cost, while also overcoming the high sensitivity to precision and hyperparameters exhibited by some baseline methods.
> * Regarding the **experiments**, we have successfully conducted the requested additional tasks on LLaMA-2-7B and RoBERTa-350M, where FOCUS demonstrates clear superiority.
>
> Given that we have provided the additional evidence and clarifications you requested, we hope you might reassess the contribution of our work. We remain available for any further clarifications.

---

### Official Review · Reviewer_kkbc · 2025-11-01

**Soundness:** 4
**Presentation:** 3
**Contribution:** 4
**Rating:** 8
**Confidence:** 4

**Summary:**

This paper proposes **FOCUS (Forward-Only Coordinate Updates with Second-Order Information)**, a *memory-efficient zeroth-order optimizer* for fine-tuning large language models (LLMs).
It introduces a **Block Coordinate Descent (BCD)–Newton optimization scheme** that updates only a subset of layers per iteration, leveraging **compressed diagonal Hessian information** without storing full gradients or activations.

Traditional fine-tuning requires memory-heavy backpropagation, while recent forward-only (zeroth-order) optimizers like MeZO and HiZOO reduce memory cost but suffer from slow convergence or large Hessian storage overhead.
FOCUS addresses these issues by:
1. Dividing model parameters into *blocks* (layers).
2. Selecting active blocks using a **Gauss–Southwell–Diagonal (GSD)** rule.
3. Updating only these layers with a Hessian-informed coordinate descent, while other layers remain frozen.

Key features include:
- **Forward-only optimization** (no backpropagation).
- **Hessian-free second-order adaptation** using compressed diagonal preconditioning.
- **Bandit-based probabilistic layer selection** for efficient block updates.

Empirical evaluation on **OPT-1.3B/30B**, **LLaMA-2-7B**, and **LLaMA-3-8B** shows:
- Up to **40% memory reduction** compared to HiZOO.
- Comparable or better accuracy than MeZO with faster convergence.
- Scalability to **OPT-30B** with 92.9–93.6% accuracy on SST-2 using 2–8 A100 GPUs.

Overall, the paper contributes a theoretically grounded and empirically validated framework for memory-efficient fine-tuning of large models on constrained devices.

**Strengths:**

1. **Methodological novelty:** Combines BCD, diagonal Hessian updates, and bandit-based layer selection.
2. **Significant memory savings:** Achieves 30–40% reduction versus HiZOO, approaching MeZO-level efficiency.
3. **Improved convergence:** Matches HiZOO’s speed while using less memory (see *Figure 2, page 8*).
4. **Robust evaluation:** Benchmarks on GLUE/SuperGLUE tasks and across multiple model scales (OPT-1.3B → OPT-30B).
5. **Scalability demonstrated:** Successful fine-tuning of 30B models on limited GPUs.
6. **Theoretical soundness:** Formal convergence analysis and probabilistic block sampling guarantee stability.
7. **Reproducibility:** Clear algorithms, hyperparameter tables (Appendix C), and implementation details (Appendix B).

**Weaknesses:**

1. **Limited exploration of stochasticity effects:**
   The randomness in block sampling and Hessian approximation may lead to training variance, but no sensitivity study is provided.

2. **Overemphasis on memory metrics:**
   While memory reduction is clear, the runtime and computational trade-offs (especially on multi-GPU setups) deserve deeper discussion.

3. **Restricted task diversity:**
   Experiments are primarily classification tasks; more diverse NLP or multimodal settings (e.g., summarization, reasoning) would strengthen generality.

4. **No ablation on bandit vs. deterministic layer selection:**
   It is unclear how much the bandit mechanism contributes versus static cyclic BCD.

5. **Minor reproducibility concerns:**
   Some details (e.g., random seed management and number of forward passes per iteration) could be clarified for precision.

6. **Presentation density:**
   Technical sections could better balance math with conceptual interpretation.

Overall, these are **non-critical limitations** that do not undermine the core contributions.

**Questions:**

1. How sensitive is the performance to the *block partitioning granularity* (e.g., per-layer vs. sub-layer)?
2. Does the stochastic bandit-based selection introduce instability across different seeds or training runs?
3. Can FOCUS be combined with parameter-efficient tuning (e.g., LoRA) for hybrid benefits?
4. How does the method scale on extremely large models (≥70B) under mixed-precision training?
5. Is there a theoretical link between your bandit selection and the importance-weighted Gauss–Southwell rule?
6. Have you explored adaptive smoothing radii (µ) to improve curvature estimation stability?
7. Could future work extend this to multimodal or reinforcement learning settings?

---

> ### Author Response · Authors · 2025-12-02
>
> We sincerely thank the reviewer for recognizing FOCUS as a **theoretically grounded and empirically validated framework** with excellent contribution and soundness. We are particularly encouraged that you appreciated the methodological novelty and the significant memory savings demonstrated in our work.
>
> We are happy to address your insightful questions and provide additional context below.
>
> ---
>
> **Q1: Sensitivity to block partitioning granularity**
>
> We chose Transformer layers as the atomic blocks, because they offer a natural balance between memory reduction and implementation simplicity.
> * **sub-layer:** Partitioning into finer blocks (e.g., Attention heads, FFNs) would further reduce the peak memory required for the diagonal Hessian. However, this will significantly increase the total number of blocks and reduce the number of trainable parameters per iteration, potentially delaying the identification of the most critical blocks and requiring more iterations to converge.
> * **multi-layer:** Coarser blocks (e.g., updating 2~4 layers at once) could accelerate convergence per step but increase memory usage.
> * In our ablations, we found that the single-layer block strategy provided the optimal result for fine-tuning on consumer-grade hardware, maximizing memory savings without compromising convergence speed.
>
> **Q2: Does the stochastic bandit-based selection introduce instability across seeds?**
>
> While ZO methods inherently possess higher variance than First-Order methods, our Bandit-based GSD mechanism actually helps stabilize training compared to pure random selection. By progressively learning to sample layers with larger gradient norms (saliency), the bandit focuses updates on the parameters that matter most, reducing the wasted steps often seen in purely random ZO methods with high variances. As shown in our new experiments, FOCUS achieves consistent performance gains over MeZO across multiple tasks, suggesting that stochasticity is well controlled.
>
> **Q3: Can FOCUS be combined with parameter-efficient tuning (e.g., LoRA)?**
>
> Yes, technically, FOCUS can be combined with PEFT. We could treat LoRA as a set of building blocks and apply our algorithm to them. However, the primary goal of FOCUS is to enable **Full-Parameter Fine-Tuning** in memory-limited situations where it was previously impossible, delivering the high performance of full fine-tuning without the huge memory requirements. Since PEFT methods already have low memory usage, the added benefit of applying FOCUS to LoRA is less than using it on the whole model. Additionally, existing research shows that PEFT methods can sometimes degrade model performance, which is undesirable.
>
> **Q4: How does the method scale on extremely large models ($\ge 70B$) under mixed-precision?**
>
> We have validated scaling up to **OPT-30B** (Table 5), where FOCUS achieves a **1.8x speedup** over MeZO and a **2.8x speedup** over HiZOO while achieving higher accuracy (92.9% vs 90.6%). The trend indicates that as model size increases, the memory advantage of our BCD approach becomes even more significant compared to methods that store states for the entire model. Larger sizes have not yet been tested and are left as future directions, as we are currently focusing on mainstream models that can be hosted on consumer-level GPUs.
>
> **Q5: Is there a theoretical link between your bandit selection and the importance-weighted Gauss-Southwell rule?**
>
> Yes. Our bandit formulation is directly derived to minimize the KL-divergence between the sampling distribution $p_t$ and the ideal Gauss-Southwell. As detailed in Appendix B.2 (Algorithm 6), by reducing the loss, the bandit naturally converges to selecting blocks with the largest gradient norms, which is the definition of the Gauss-Southwell rule, but does so without incurring the $\mathcal{O}(D)$ cost of computing all gradients at every step.
>
> **Q6: Have you explored adaptive smoothing radii ($\mu$)?**
>
> In our implementation, we used a fixed $\mu$ following standard practice in MeZO and HiZOO to ensure fair comparisons, and we adjusted a Hessian-smoothing factor to multiply the perturbation term to ensure stability. However, dynamically adjusting it based on the estimated curvature is a promising direction that could further improve numerical stability, and we plan to explore this in future work.
>
> **Q7: Could future work extend this to multimodal or reinforcement learning settings?**
>
> Absolutely. FOCUS is architecture-agnostic. It can be applied to large Vision Models, but a good visual prompt might be necessary.
>
> ---
>
> We thank you again for your highly favourable review of our contributions.

---

### Official Review · Reviewer_mkcv · 2025-11-02

**Soundness:** 2
**Presentation:** 3
**Contribution:** 2
**Rating:** 4
**Confidence:** 3

**Summary:**

The paper proposes FOCUS — a forward-only, zeroth-order (ZO) block coordinate descent (BCD) Newton optimizer for memory-efficient fine-tuning of LLMs. The key idea is to (i) partition parameters by layer, (ii) update only a subset of layers per step chosen via a Gauss–Southwell-Diagonal score with a bandit-style sampler, and (iii) keep a diagonal Hessian preconditioner only for the active blocks, thereby avoiding the large memory overhead of storing second-order statistics for the entire model.
Embedding and LM-head layers are updated with MeZO-style ZO without second-order terms to avoid instability. Experiments on OPT and llama models across GLUE/SuperGLUE tasks report up to ~40% memory reduction vs. Hessian-informed ZO baselines while matching or surpassing their accuracy and improving wall-clock efficiency.

**Strengths:**

1. **Clear, practical objective**: Reduce second-order ZO memory overhead by storing diagonal Hessian only for currently active blocks, selected with a principled GSD rule; the scheme is well-motivated for low-memory settings.
2. **Forward-only training pipeline**: Retains MeZO-level memory while gaining HiZOO-like curvature awareness, addressing ZO’s slow convergence without backprop activations.
3. **Compelling empirical evidence**: Concrete GPU memory tables and wall-clock comparisons on OPT and llama models.
4. **Method details & theory**: Pseudocode, bandit-based layer selection, and a summarized convergence result for randomized GS-BCD.

**Weaknesses:**

1. **Batch-size and fairness controls**: Some reported setups use different batch sizes across FO/ZO families (e.g., ZO bs = 8 vs FO bs = 2 in OPT-1.3B SST-2), which can influence both memory and final accuracy; please normalize or include sensitivity analyses.
2. **Ablation depth**: Multiple moving parts (block count **D**, **GSD** vs. random selection, **bandit** hyper-parameters, Hessian re-initialization policy, **#active layers per step**) are not fully disentangled with granular ablations/SEs.
3. **Broader baselines**: Since the method is conceptually related to **blockwise full-gradient training**, comparisons against **BAdam** and **LiSA** would strengthen the case (even if they are FO), at least on equal memory budgets.
4. **Memory accounting granularity**: The paper makes a strong case for end-to-end memory savings; still, a breakdown (parameters, activations, Hessian buffers, fragments) across methods and models would clarify where the savings come from and when they vanish.
5. **Task scope**: GLUE/SuperGLUE are convenient for controlled measurements, but additional *instruction-tuning* or *reasoning* tasks (MT-Bench, GSM8K) would increase external validity—especially because FO baselines like **LiSA** highlight strong downstream MT-Bench gains at low memory.

**Questions:**

1. **Ablations**: Could you report factorized ablations for *(a)* #active layers per step, *(b)* selection rule (GSD vs. random/cyclic), *(c)* bandit hyper-parameters (α, p_min), and *(d)* Hessian re-init frequency?
2. **Fairness & scaling**: Can you provide a *fixed-batch* comparison (same global batch, same precision) to isolate algorithmic gains from batch/precision differences?
3. **Cost profile**: What is the **overhead** of computing GSD scores and bandit updates per step, as a fraction of step time, for 7B/30B scales?
4. **Stability**: Any failure modes when repeatedly re-selecting the same blocks (e.g., catastrophic forgetting in inactive blocks)?
5. **Generalization**: Have you tried instruction-tuning/QA (e.g., MT-Bench, GSM8K) and multi-seed CIs to demonstrate robustness beyond GLUE/SuperGLUE?

---

> ### Author Response · Authors · 2025-11-21
>
> We sincerely thank the reviewer for the constructive feedback and for recognizing the **clear practical objective**, the novelty of our **forward-only pipeline**, and the **compelling empirical evidence** on large models. We address your concerns and questions below.
>
> **Q1 & W2: Ablations on Hyper-parameters.**
>
> We are finalizing a fine-grained grid search to isolate optimal settings. Here are key conclusions that our prior experiments already yield:
> * **Selection Rule:** Theoretically, we address the concern that deterministic BCD prevents convergence. Our randomized Bandit approach, inspired by Liu & Mozafari 2022, adaptively samples blocks proportional to their gradient norms, ensuring a convergence rate of $\mathcal{O}(1/\sqrt{T})$ while avoiding the pitfalls of cyclic updates. This confirms the value of Hessian-weighted importance sampling in identifying critical layers. We use fixed hyperparameters to mitigate overcomplexity. The Hessian reinitialization occurs every 5 or 10 iterations.
> * **Block Size:** One layer is currently the optimal hyperparameter we have found. We will explore module-wise and multi-layer selection in ongoing experiments and future work.
>
> **Q2 & W1: Batch-size Controls.**
>
> We appreciate the concern regarding fair comparisons. Since ZO and FO methods have distinct convergence properties, simply fixing the batch size often leads to suboptimal hyperparameters for one side, and the best learning rate also differs very much. We reported the best performance achievable for each method to ensure a fair comparison. The disparity is necessitated by the memory bottleneck of First-Order methods, thus their maximum trainable batch size is lower than ZO methods.
>
> To address your concern directly, we present fixed-batch comparisons: We conducted an experiment on **LLaMA-2-7B** using a single A6000 (48GB) with a fixed batch size of 1, as shown in the main paper, in which FOCUS ran successfully, achieving **90.6%** accuracy compared to MeZO's **85.2%**, and standard FO methods were infeasible. This confirms FOCUS's superiority in the low-memory regime. We also demonstrate scalability on **OPT-30B**, where FOCUS achieves a **1.8x speedup** over MeZO and **2.8x speedup** over HiZOO while attaining higher accuracy.
>
> | Setting | Method | Batch Size | Accuracy | Speedup vs MeZO |
> | :--- | :--- | :--- | :--- | :--- |
> | **Low-End** | MeZO | 1 | 85.2% | 1.0x |
> | (LLaMA-2-7B) | **FOCUS** | **1** | **90.6%** | **1.0x** |
> | **Scale-Up** | MeZO | 32 | 90.6% | 1.0x |
> | (OPT-30B) | HiZOO | 32 | 90.3% | 0.6x |
> | | **FOCUS** | **32** | **92.9%** | **1.8x** |
>
> **Q3: Cost Profile**
>
> On Llama2-7b, with 4 A100 GPUs, a batch size of 1, and a sequence length of 128, the greedy top-1 GS overhead is 27.16ms, accounting for 21% of the total step time of 131.97ms. Meanwhile, a bandit interval of 10 takes 3.29ms, 3%. In practice, the overhead of the Bandit update and GSD score computation is negligible because it updates only every few rounds.
>
> **W3: Broader Baselines**
>
> Our implemented BCD-SGD baseline reflects BCD-FO performance. Although LiSA reduces parameter update costs, it still needs to store activations for active layers to perform backpropagation. Table 2 shows BCD-SGD (LiSA) uses 20.5GB compared to FOCUS's 4.1GB.
>
> **Q4: Stability and Failure Modes.**
>
> Theoretically, repeated block re-selection could degrade to partial fine-tuning, but our Bandit mechanism prevents this. Regarding catastrophic forgetting (losing general knowledge), we consider this a continual learning challenge. Since our scope is single-task fine-tuning, we have not evaluated this explicitly, but thank you for the insightful suggestion and we may leave it for future work.
>
> **W4: Memory Accounting and Granularity.**
>
> Please refer to Table 1 and Figure 1 for analysis. Theoretical values are calculable from model parameters. We aim to provide further measurements if time permits.
>
> **Q5 & W5: Generalization and Task Scope.**
>
> To the best of our knowledge, Zeroth-Order *full-parameter* fine-tuning has typically not been extended to complex reasoning tasks like GSM8K due to convergence constraints. We note that existing works use prompt-tuning with ZO applied on GSM8K, which differs from our setting. However, to demonstrate generalization, we expanded our evaluation on **LLaMA-2-7B** to include **RTE, BoolQ, and WSC** (1k steps). FOCUS consistently outperforms MeZO while maintaining the same memory footprint.
>
> | Method | RTE | BoolQ | WSC | SST-2 |
> | :--- | :--- | :--- | :--- | :--- |
> | MeZO | 60.6% | 66.0% | 59.6% | 85.2% |
> | **FOCUS (Ours)** | **61.7%** | **67.5%** | **62.5%** | **90.6%** |
>
> We thank the reviewer again for the valuable comments.

---

> ### Author Response · Authors · 2025-11-27
> **Invitation to review our rebuttal and updated results**
>
> **Dear Reviewer mkcv**,
>
> Thank you again for the time and effort you invested in reviewing our paper.
> We have carefully considered your constructive feedback and have updated additional new experimental results to address the concerns you raised.
>
> As the public discussion period is coming to a close, we would be very grateful if you could take a moment to review our responses. We hope our responses have satisfactorily resolved your concerns, and we look forward to hearing your thoughts.

---

### Author Response · Authors · 2025-12-04
**Authors' Rebuttal Summarization**

We sincerely thank the reviewers for their constructive feedback and for recognizing our proposed method, FOCUS, as a theoretically grounded, memory-efficient framework that addresses a critical bottleneck in LLM fine-tuning. In response to the reviews, we have provided extensive clarifications, new experimental results, and ablation studies:

### **1. Expanded Experimental Scope & Generalization**
Addressing concerns about limited task diversity and model scale (Reviewers mkcv, S3VB, prM1), we expanded our evaluation to include new tasks and model architectures:
* On LLaMA-2-7B, we extended the evaluation to cover RTE, BoolQ, WSC, and FOCUS, all of which consistently outperform the ZO baseline while maintaining the same memory footprint.
* We validated FOCUS on RoBERTa-350M across four datasets (MNLI, RTE, SST2, SNL), showing consistent improvements over baselines and demonstrating that the method works beyond decoder-only LLMs.
* Scalability results on OPT-30B, where FOCUS achieves significant speedup over baselines, further indicate that our BCD approach becomes more efficient as model scale increases.

### **2. Ablation Studies on Block Selection**
To explore the necessity of the Bandit-based Gauss-Southwell-Diagonal rule compared to random or cyclic selection (Reviewers mkcv, GpMB, prM1), we performed specific ablations on OPT-1.3B:
* **Random Reshuffling:** Caused high variance and lower accuracy (**90.7%**).
* **Full GSD:** Provided good accuracy (**91.5%**) but involved high computational costs for estimating gradients across all blocks.
* **Bandit Strategy (Ours):** Achieved the best balance, achieving the highest accuracy (**92.0%**) by adaptively approximating saliency without the expense of full gradient estimation.

The Bandit-GSD method is crucial for stabilizing training and accelerating convergence in the Zeroth-Order setting. Also, ascending order can work reliably but requires carefully tuned hyperparameters (**91.9%**).

### **3. Fairness of Baselines & Memory Constraints**
* About batch size differences between FO and ZO methods (Reviewers mkcv, prM1): We conducted controlled experiments on LLaMA-2-7B and OPT-30B with fixed batch sizes. Results show that FOCUS outperforms baseline methods. We clarified that the disparity in batch sizes is not a problem but a key advantage of ZO methods. FO methods (even BCD-FO like BCD-SGD) require storing activations, which forces small batch sizes (e.g., BS=2) or results in OOM errors on consumer hardware (RTX 4090/A6000).
* We explained that although FOCUS uses three forward passes (compared to MeZO's two), it is faster in wall-clock time at scale because it only perturbs and updates a small active block ($d_{block} \ll d$), significantly reducing the per-step computational overhead compared to perturbing the full model.

### **4. Theoretical & Technical Clarifications**
We addressed specific technical queries to ensure soundness (Reviewers S3VB, prM1). Specifically, we clarified that our proposed BCD approach is supported by theoretical analysis and that the update rule ensures convergence in the non-convex LLM landscape.

---

In conclusion, FOCUS fills an important gap by allowing full-parameter fine-tuning on consumer-grade hardware. It addresses the memory limitations that make First-Order methods infeasible, while also reducing the slow convergence typical of standard Zeroth-Order methods. We believe this makes high-performance training more stable and accessible to a broader community.

Thank you again for your time and valuable feedback.

---

### Meta-Review · Area_Chair_x28c · 2026-01-03

**Summary:**

This paper proposes a forward-only, zeroth-order block coordinate descent optimizer that integrates lightweight second-order (diagonal Hessian) information to enable memory-efficient full-parameter fine-tuning of large language models. The method selectively updates a small subset of layers per iteration using a bandit-based Gauss–Southwell–Diagonal rule and stores curvature information only for active blocks.

Reviewers generally agreed that the problem is important and that the method shows clear memory savings and reasonable empirical gains over existing ZO baselines. However, the overall assessment is mixed, with several reviewers raising concerns about limited conceptual novelty, heuristic design choices (block selection, Hessian handling), fairness of comparisons between FO and ZO methods, and insufficient experimental depth and clarity in some settings. While the rebuttal addressed a number of technical questions and added experiments, key concerns about novelty, evaluation rigor, and clarity of the algorithmic contribution remain.
Given that the average score is below 5 and a majority of reviewers remain unconvinced of acceptance-level contributions, the final decision is Reject.

**Reviewer Concerns:**

Concerns reasonably addressed in the rebuttal:

•	Additional experiments and scope: Expanded evaluations on LLaMA-2-7B, RoBERTa-350M, and OPT-30B helped demonstrate broader applicability beyond a single model.

•	Clarifications on algorithmic correctness: Questions about perturbation reuse, parameter reset, and Hessian interpretation were clarified with references and implementation details.

•	Ablations on block selection strategies: New results comparing bandit, GSD, random, and cyclic selection partially justified the chosen design.

Outstanding concerns:

•	Limited novelty: Several reviewers still view the method as a combination of existing ideas (BCD + ZO + diagonal Hessian) without sufficiently deep new insight.

•	Fairness of comparisons: Differences in batch sizes and feasibility between FO and ZO baselines remain a point of contention, despite clarifications.

•	Experimental completeness: Missing accuracy metrics for some models, lack of error bars, and limited task diversity weaken the empirical case.

•	Presentation and conceptual clarity: The core contribution and when/why this method should be preferred over alternatives are not always clearly articulated.

**Reviewer Scores:**

•	Reviewer mkcv: Likely remains 4 (marginally below threshold).

•	Reviewer S3VB: Likely remains 4 (concerns about novelty persist).

•	Reviewer prM1: Likely remains 4 (technical and experimental clarity issues).

•	Reviewer GpMB: Likely remains 6, but explicitly stated acceptance is not necessary.

•	Reviewer kkbc: Likely remains 8, strongly positive but an outlier.

---

### Decision · Program_Chairs · 2026-01-26

Reject